# A novel esterase regulates *Klebsiella pneumoniae* hypermucoviscosity and virulence

Lijun Wang[1,2☯], Zhe Wang[3☯], Hua Zhang[4☯], Qian Jin[2], Shuaihua Fan[5], Yanni Liu[2], Xueting Huang[2], Jun Guo[6], Chao Cai[3]*, Jing-Ren Zhang[2]*, Hui Wu[4]*

**1** Department of Laboratory Medicine, Beijing Xiaotangshan Hospital, Beijing, China, **2** Center for Infectious Disease Research, School of Basic Medical Sciences, Tsinghua Medicine, Tsinghua University, Beijing, China, **3** Shandong Key Laboratory of Glycoscience and Glycotherapeutics, School of Medicine and Pharmacy, Ocean University of China, Qingdao, China, **4** Oregon Health and Science University School of Dentistry, Portland, Oregon, United States of America, **5** Tsinghua Medicine, Tsinghua University, Department of Respiratory and Critical Care Medicine, Beijing Tsinghua Changgung Hospital, Beijing, China, **6** Department of Geriatric Medicine, Beijing Tsinghua Changgung Hospital, Tsinghua Medicine, Tsinghua University, Beijing, China

☯ These authors contributed equally to this work.
* caic@ouc.edu.cn (CC); zhanglab@tsinghua.edu.cn (J-RZ); wuhu@ohsu.edu (HW)

**Data Availability Statement:** Raw RNA-seq data are available in NCBI's GEO database (accession number GSE205098). All data are available in the main text or the supplementary materials.

## Abstract

*Klebsiella pneumoniae*, an emerging multidrug-resistant pathogen, exhibits hypermucoviscosity (HMV) as a critical virulence trait mediated by its capsular polysaccharide (CPS). Recent discoveries have determined acetylation as a significant modification for CPS, although its impact on HMV and virulence was previously unknown. This study elucidates the roles of two enzymes: ***K**lebsiella **p**neumoniae* **A**cetylated **C**PS **E**sterase (KpACE), an esterase that removes acetyl groups from CPS, and WcsU, an acetyltransferase that adds acetyl groups to CPS. KpACE is highly upregulated in an *ompR*-deficient mutant lacking HMV, and its overexpression consistently reduces HMV and diminishes virulence in a mouse model of pneumonia. The esterase domain-containing KpACE effectively deacetylates model sugar substrates and CPS-K2. Site-directed mutagenesis of the conserved catalytic histidine residue at position 370 significantly reduces its enzymatic activity. This reduction correlates with decreased HMV, affecting key virulence traits including biofilm formation and serum resistance. Similarly, a deficiency in the *wcsU* gene abolishes CPS acetylation, and reduces HMV and virulence. These results highlight the importance of the delicate balance between CPS acetylation by WcsU and deacetylation by KpACE in regulating the pathogenicity of *K. pneumoniae*. Understanding this balance provides new insights into the modulation of virulence traits and potential therapeutic targets for combating *K. pneumoniae* infections.

**Funding:** This work was supported by High-Level Public Health Technical Talent Construction Project of Beijing Municipal Health Commission No. Disciplinary Backbone-02-48 (LW), Weight Endowed Professorship from Oregon Health & Science University School of Dentistry (HW), NIH T90DE030859 (HW) and National Natural Science Foundation of China 82330071 (J-RZ). The funders had no role in study design, data collection and analysis, decision to publish, or preparation of the manuscript.

**Competing interests:** The authors have declared that no competing interests exist.

## Author summary

*Klebsiella pneumoniae* is an antibiotic-resistant pathogen known for its virulence factor, hypermucoviscosity (HMV), which relies on the production of capsular polysaccharides (CPS). This study highlights the roles of two key enzymes: KpACE, an esterase that removes acetyl groups from CPS, and WcsU, an acetyltransferase that adds acetyl groups to CPS. Overexpression of KpACE reduces HMV and decreases virulence in a mouse pneumonia model by deacetylating CPS. Conversely, a deficiency in the *wcsU* gene, which is responsible for acetylating CPS, also results in decreased HMV and virulence. The balance between CPS acetylation by WcsU and deacetylation by KpACE is crucial in regulating *K. pneumoniae* pathogenicity. This balance affects various virulence factors, including biofilm formation and serum resistance, highlighting the importance of CPS acetylation dynamics in the pathogen's fitness and virulence. Understanding these mechanisms offers insights into potential therapeutic targets to mitigate the virulence and antibiotic resistance of *K. pneumoniae*.

## Introduction

*Klebsiella pneumoniae* has emerged as a prominent pathogen causing severe infections [1]. The clinical management of its hypervirulent strains poses particular challenges, due to its high mortality rate. The emergence of multidrug-resistant hypervirulent *K. pneumoniae* strains further exacerbates these challenges [2]. Previous studies have indicated a close correlation between the hypermucoviscosity (HMV) phenotype of *K. pneumoniae* and its virulence [3,4]. The formation of the HMV requires the synthesis of capsule polysaccharides (CPS), but the regulatory mechanisms governing the formation of the HMV, particularly the role of CPS modifications in *K. pneumoniae* virulence and pathogenesis, are unknown.

CPS is essential in determining the HMV phenotype and virulence. The structure and rheological properties of CPS have been recognized as key factors in mediating CPS activities. Specifically, CPS exhibits acetyl substituents in certain *K. pneumoniae* serotypes like K1 and K2, which imparts a flexible and viscous character to the bacterial surface [5–8]. Additionally, the acetylation of CPS has been associated with various pathogenic traits of *K. pneumoniae*, including cytokine production, influencing bacterial capability to evade immune responses [5].

The process of *O*-acetylation, facilitated by the transfer of acetyl residues to carbohydrate backbones by acetyltransferases, and its counterpart, *de-O*-acetylation, mediated by acetylesterases or other hydrolytic enzymes, finely regulate acetylation levels [9, 10]. The balanced action of these processes is critical, and acetylesterase, found in several bacteria, play key roles in diverse biochemical pathways. For example, NeuA, a new class of Sialyl *O*-acetylesterase, modulates the acetyl modification of CPS in *E. coli*. NeuA is a bifunctional enzyme, including N-terminal cytidine 5'-monophospho-N-acetylneuraminic acid (CMP-Neu5Ac) synthetase and C-terminal esterase, mediating the acetylation of monomeric sialic acids [11]. A homologous bifunctional NeuA esterase from Group B Streptococcus modulates the acetylation of Sias on its cell surface [12] and influences the host-pathogen interplay [13], highlighting the importance of the acetylation dynamics and their modification of bacterial surface constitutes.

In *K. pneumoniae*, it is well documented that RmpADC mediates the abundance and length of CPS [14,15], and other unrecognized molecules mediated by OmpR are involved in the HMV formation and the pathogenesis [16]. In this study, we investigated the role of OmpR-

regulated factors affecting HMV formation through transcriptomics and identified a gene encoding a putative esterase named KpACE (*K. pneumoniae* acetylated capsule esterase), which was highly upregulated in the *ompR* mutant lacking HMV. KpACE overexpression reduces mucoviscosity and virulence by removing acetyl groups from modified carbohydrates and CPS of K2-serotype (CPS-K2). Additionally, we uncovered a gene, *wcsU*, encoding a putative acetyltransferase in the *cps* synthesis locus, which mediates addition of acetyl residues to CPS-K2.

Our studies demonstrate that *K. pneumoniae* employs KpACE and WcsU to control acetyl group levels in CPS-K2, influencing mucoviscosity and virulence. Overexpression of KpACE and deletion of WcsU both led to the attenuation of *K. pneumoniae* virulence, implying their potential utility in combating *K. pneumoniae* infections. Our findings unveil a previously unknown deacetylation mechanism, providing new insights into exploring the acetylation-mediated HMV formation for novel therapeutics.

## Results

### Identification of a novel acetylesterase, KpACE, that inhibits HMV and virulence

We have determined the essential role of OmpR in both the HMV formation and virulence of hypervirulent strains independently of recognized HMV determinants such as RmpD [16]. To comprehensively investigate the molecular mechanisms involved in the HMV formation, we performed an additional RNA-seq analysis of *K. pneumoniae* grown at log-phase to identify the OmpR regulon associated with HMV. Twenty-three differentially expressed genes (DEGs) were identified between the wild-type and the Δ*ompR* mutant (S1 Table). Gene Ontology (GO) enrichment analysis of these DEGs identified the cellular processes associated with mucoviscosity, and with the transport category being the most frequently represented GO term (S1 Fig). Notably, the most upregulated genes were enriched in the GO term of organic substance transport. Because the down-regulated genes in the Δ*ompR* mutant had no impact on HMV formation based on a previous study of Tn-seq library screening [17], we focused our investigation on the upregulated genes due to their potential involvement in mediating mucoviscosity. Among the top three upregulated genes, VK055_3347 and VK055_4943 were annotated as putative carbohydrate porins, hinting at their potential association with CPS-mediated mucoviscosity formation. Thus, we investigated them by genetically engineering the wild-type strain to overexpress either the VK055_3347 locus or VK055_4943. Unfortunately, the construction of a plasmid overexpressing VK055_4937 (*mglA*), another top-upregulated gene, failed several times for unknown reasons, so its impact on mucoviscosity could not be further studied. Our current study thus focuses on investigating the VK055_3347 locus.

Genomic analysis suggested that the VK055_3347, VK055_3348 (*kpACE*), and VK055_3349 constitute a gene locus (Fig 1A), so we investigated their individual and collective effects on mucoviscosity. qRT-PCR data demonstrated that the transcriptional levels of VK055_3347, *kpACE*, VK055_3349, and VK055_4943 were significantly upregulated (Fig 1B). The sedimentation-assay results showed that overexpression of the VK055_3347 locus (WTp3347 locus, TH16330), but not VK055_4943 (WTp4943, TH16343), decreased mucoviscosity compared to the wild-type vector control (Fig 1C). Interestingly, overexpression of the sole gene *kpACE* located in the middle of the locus (WTpkpACE, TH16356) was sufficient to reduce mucoviscosity, emphasizing the importance of *kpACE*. An electrophoretic mobility shift assay (EMSA) revealed that recombinant OmpR (rOmpR) directly bound to the promoter region of VK055_3347 locus ($P_{VK055\_3347}$) (S2 Fig), indicating that OmpR represses the expression of the *kpACE* directly.

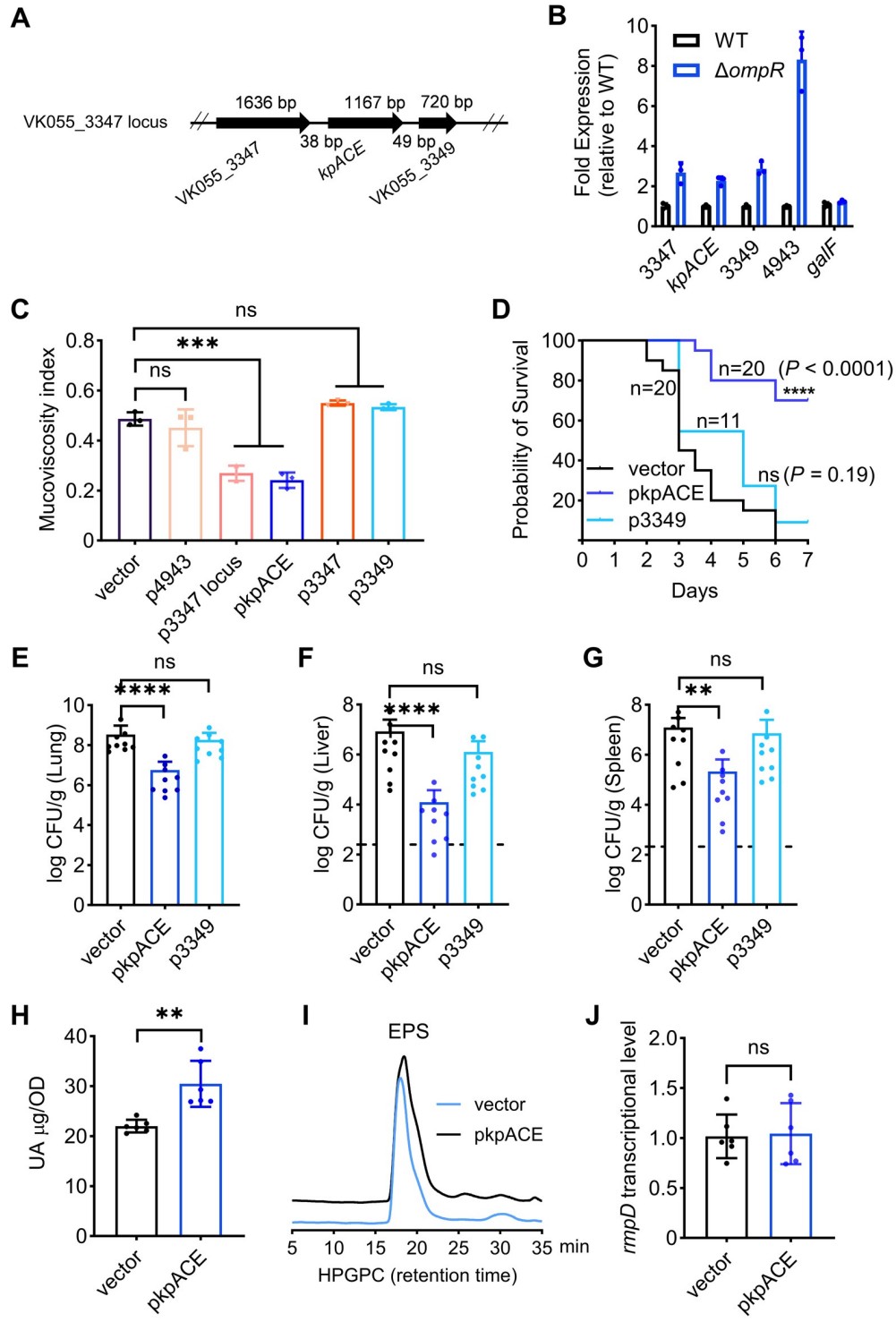

**Fig 1. The *kpACE* overexpression reduces mucoviscosity. (A)** Diagram of the VK055_3347 locus (not drawn to scale). **(B)** The transcriptional expression of genes. The expressional levels were quantified by qRT-PCR, and the levels in the Δ*ompR* mutant were normalized to these of wild-type (WT). The *galF* gene was used as a negative control. **(C)** Mucoviscosity of saturated overnight cultures of ATCC43816 derivative carrying plasmid expressing VK055_4943 (p4943), VK055_3347 locus (p3347 locus), VK055_3347 (p3347), *kpACE* (pkpACE), and VK055_3349 (p3349), respectively. **(D-G)** Effects of *kpACE* overexpression on virulence in a pneumonia model using both female and male mice. (D) Survival rates of mice infected by *K. pneumoniae* variants. After infecting mice with 2,000 CFU of various ATCC43816 recombinant strains, including WT-vector (14 female and 6 male mice), pkpACE (14 female and 6 male

mice), and p3349 (6 female and 5 male mice), the survival rates were monitored for 7 days post-infection. The Gehan-Breslow-Wilcoxon test was performed to compare the survival rates. Bacterial burdens in lungs (E), livers (F), and spleens (G) of infected mice (6 female and 3 male) were determined 48 h post intranasal inoculation. Each dot represents one mouse, and dotted lines represent the detection limit. One-way ANOVA with Dunnett's multiple comparisons test was performed to determine the statistical significance. The separate results for female and male mice were presented in S11 Fig. **(H)** The uronic acid contents of capsules. **(I)** The size of exopolysaccharides (EPS). 10 mg of purified EPS from both the overexpressed KpACE strain (pkpACE) and wild-type (vector) were analyzed using the high-performance gel permeation chromatography (HPGPC), which separates EPS based on their size. Retention times are used to compare the relative sizes of different molecules. If two EPS have similar retention times, it suggests they have similar molecular weights or sizes. **(J)** The *rmpD* transcriptional levels determined by RT-qPCR. Data are presented as mean ± SD. The unpair *t-test* was then performed to determine statistically significant differences between each isogenic strain and its vector control.

We then evaluated virulence properties of this overexpression strain using a mouse model of pneumonia. The survival rate of mice infected with *K. pneumoniae* overexpressing wild-type KpACE was significantly enhanced (Fig 1D). Additionally, the dissemination of bacteria to different organs was assessed post-infection. The control wild-type bacteria showed high colonization in the primary lung lesion (approximately $10^8$ CFU/g) and metastatic sites (liver and spleen, approximately over $10^6$ CFU/g), whereas the WTpkpACE strain exhibited lower bacterial burdens in the primary lung lesion ($< 10^7$ CFU/g) (Fig 1E) and even more significant reduction in the metastatic sites of the liver ($< 10^4$ CFU/g) (Fig 1F) and spleen ($< 10^5$ CFU/g) (Fig 1G). In contrast, an isogenic strain that overexpressed VK055_3349 did not alter mucoviscosity, and retained virulence comparable to its wide-type counterpart, suggesting that KpACE selectively attenuated virulence. To further explore the role of *kpACE* in virulence, we generated an isogenic *kpACE* mutant (TH16429) and evaluated its virulence profile. Interestingly, the mutant exhibited the same virulence profile as the wild-type, as shown in S3 Fig. These data suggest that the protective effect of KpACE we observed is dependent on its elevated expression levels, and the role of KpACE is complex and warrants further investigation.

To understand the negative regulation of KpACE on mucoviscosity, we assessed changes in capsule and RmpD levels, two well-documented determinants of mucoviscosity [14, 18]. Surprisingly, despite lower mucoviscosity, the WTpkpACE strain produced more capsules than the wild-type, as indicated by higher uronic acid contents (Fig 1H). The size of exopolysaccharides (EPS) was also evaluated using the high-performance gel permeation chromatography (HPGPC) [19]. The peaks and retention time for the EPS from both strains were similar (Fig 1I), indicating that their molecular weights are comparable. These results suggest that the overexpression of KpACE does not alter the length of polysaccharide. Moreover, the transcriptional levels of *rmpD* did not change significantly (Fig 1J). These intriguing findings suggest that the negative regulation of mucoviscosity and virulence by KpACE might involve additional, as yet unidentified molecular mechanisms.

To further investigate the interplay between KpACE and mucoviscosity, we characterized this gene product. A search of the UniProt database revealed that KpACE is predicted as an Endo-1,4-β-xylanase *in silico*. Endo-1,4-β-xylanase (EC 3.2.1.8) is a glycoside hydrolase enzyme capable of degrading a polysaccharide-arabinoxylan [20]. Additionally, we analyzed the conservation of KpACE across 1,819 publicly available complete genomes of *K. pneumoniae*. KpACE homologues were identified in 82.9% (n = 1503) of these genomes, with a remarkably high sequence identity of 99.88% (S1 Data). This protein is not only prevalent but also frequently found across a variety of K-serotype strains, such as KL64, KL47, KL2, KL1, and others (S4A Fig and S2 Table). The multisequence alignment of KpACE homologues from ten representative strains of *K. pneumoniae* as depicted in S4B Fig, highlights a high degree of

conservation, with only a few variations. These results underscore the importance of defining the function of this conserved KpACE protein.

## Structure analysis of KpACE

The lack of functional illustration of KpACE in the literature prompted us to characterize this protein further. The *kpACE* gene encodes 388 amino acids with a predicted molecular weight of 43.1 kDa. KpACE is composed of a signal peptide (1–21 aa), a family 48 carbohydrate-binding module (CBM48) (42–102 aa), and an esterase catalytic domain (130–388 aa) (S5A Fig). It was classified into an Esterase-like family (IPR000801), indicating its potential role in removing acetylation from saccharides [21].

To determine if KpACE belongs to any known carbohydrate esterase (CEs) family, we also queried the genome sequence of *K. pneumoniae* ATCC43816 using the Carbohydrate-active enzymes database (CAZy) (http://www.cazy.org/b3607.html). The CAZy database documented 110 potential carbohydrate-active enzymes from the *K. pneumoniae* ATCC43816 genome, but surprisingly, KpACE was not included, suggesting that its unique function is yet to be defined.

To gain insights into its structural properties, we utilized AlphaFold to predict its three-dimensional structure [22]. The predicted model of KpACE showed high confidence with a predicted template modeling (pTM) score of 0.9107 and a per-residue accuracy score (predicted local-distance difference test, pLDDT) of 94.76 (S5B Fig). Notably, three classic His-Ser-Asp catalytic triads for hydrolysis were identified *in silico*, namely H180-S185-D217, H291-S287-D248, and H370-S278-D339. To further explore its function, we performed a structural similarity search using the Dali Server [23]. The homologues of KpACE were primarily identified with the CEs family, and the closest homologue was DmCE1B_ct (PDB:7b6b) from *Dysgonomonas mossii* [24] with a *Z* score of 38.3, sequence identity of 32%, and a root mean square deviation (RMSD) of 1.7 (S5C Fig). Among the three predicted catalytic triads, the His180 residue is conserved between KpACE and DmCE1B_ct (S5D Fig), and the catalytic triad H370-S278-D339 of KpACE is conserved as well (S5E and S5F Fig). However, the histidine residue at position 291 is not conserved among KpACE homologues (as shown in S5F Fig), and thus not pursued further experimentally. This alignment analysis suggests the potential role of the conserved H370 and H180 residues.

## The catalytic activity of KpACE

Accordingly, two variants of pkpACE with a single amino acid mutation (pkpACE$^{H180A}$ or pkpACE$^{H370A}$) were generated to investigate their activity. Overexpression of kpACE$^{H370A}$ (WTpkpACE$^{H370A}$, TH16541) was no longer able to reduce high mucoviscosity exhibited by the wild-type KpACE, indicating that the H370 mutation killed KpACE's ability to reduce mucoviscosity. In contrast, overexpression of kpACE$^{H180A}$ (WTpkpACE$^{H180A}$, TH16539) did not alter its ability to decrease mucoviscosity (Fig 2A), suggesting that the residue H180 is not important for its activity. Together, these data indicate that the H370 residue is crucial for KpACE's activity in decreasing mucoviscosity.

As the KpACE protein shares significant structural homology with DmCE1B_ct, and DmCE1B_ct acts on acetylesterase substrates [24], we investigated whether the KpACE protein possesses a similar substrate specificity. A model acetyl ester substrate, 4-methylumbelliferyl acetate (4-MU-Ac), was used to determine the enzymatic activity of recombinant KpACE (rKpACE) protein. The rKpACE protein (S6A Fig) was active over a broad pH range from 6.8 to 8.0, and the highest activity was observed at pH 7.4 (Fig 2B). Additionally, rKpACE exhibited strong catalytic activity, with a $V_{max}$ of 1.7 nmol$^{-1}$ min$^{-1}$ mg$^{-1}$ and a $K_m$ of 0.16 ± 0.03 mM

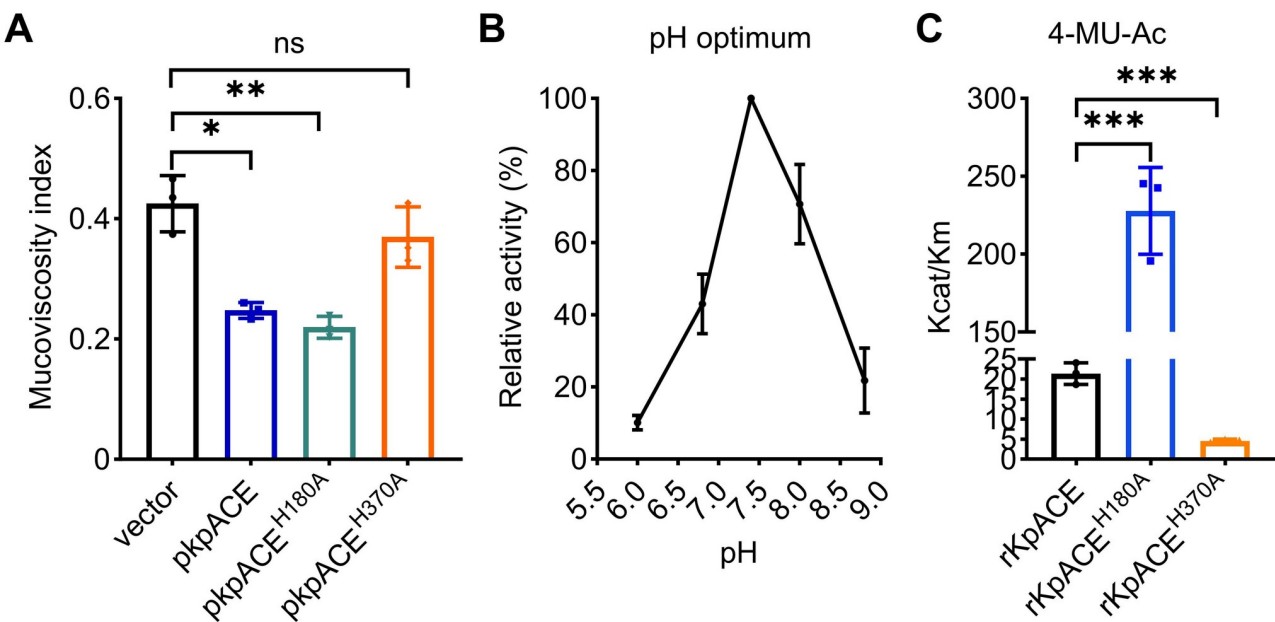

**Fig 2. Enzymatic activity of KpACE and its site-directed mutant variants.** (**A**) Mucoviscosity of overnight cultures of ATCC43816 derivatives carrying plasmids expressing KpACE, KpACE[H180A], and KpACE[H370A], respectively. (**B**) Optimal pH for KpACE activity. pH profile was determined by measuring esterase activity at pH 6.0, 6.8, 7.4, 8, and 8.8, respectively, using 1 mM of 4-methylumbelliferyl acetate (4-MU-Ac) as a model substrate. 100% represents the maximal activity determined for rKpACE. (**C**) Kinetic parameters of KpACE. The catalytic activity of rKpACE, rKpACE[H180A], and rKpACE[H370A] was determined using 4-MU-Ac at pH 7.4. The data were nonlinearly fitted to the Michaelis-Menten equation by GraphPad Prism v9.3.1. Data are presented as mean ± SD from three biological replicates.

(S6B Fig). Additionally, rKpACE exhibited catalytic activity on acetylated monosaccharides, such as fully acetylated xylose and galactose (S6C Fig). To determine the role of the conserved enzymatic residue H180 and H370, we expressed and purified protein variants of both rKpACE[H180A] and rKpACE[H370A]. As predicted, the rKpACE[H370A] had minimal acetylesterase activity (Fig 2C), suggesting that the catalytic domain is crucial for its enzymatic activity. Surprisingly, the rKpACE[H180A] variant showed a dramatic increase in its enzymatic efficiency, about 10-fold higher than that of rKpACE (Fig 2C). These data suggest that KpACE is an active carbohydrate acetylesterase with broad substrate specificity, and the residue H370 is essential for its catalytic activity.

## KpACE deficiency increases CPS acetylation and WcsU deficiency eliminates CPS acetylation

KpACE is predicted to possess a signal peptide domain, suggesting it is either a periplasmic or an extracellular enzyme. As shown in S7 Fig, the protein KpACE was detected in the culture supernatant, indicating it is secreted outside of bacterial cells. Intriguingly, the molecular weight of KpACE in the supernatant was slightly larger than that detected in the cell pellet. This discrepancy suggests that KpACE might undergo post-translational modifications or interact with other unknown cellular components during secretion.

Given its extracellular localization, we speculated that the rKpACE could directly act on mucoviscosity *in vitro*. When *K. pneumoniae* was grown in the presence of either rKpACE or rKpACEH[180A], the mucoviscosity was lower than that of *K. pneumoniae* grown in the presence of a dead enzyme, rKpACEH[370A] (Fig 3A). These data indicate that rKpACE enzymatic activity is responsible for its ability to reduce mucoviscosity. Furthermore, we examined the

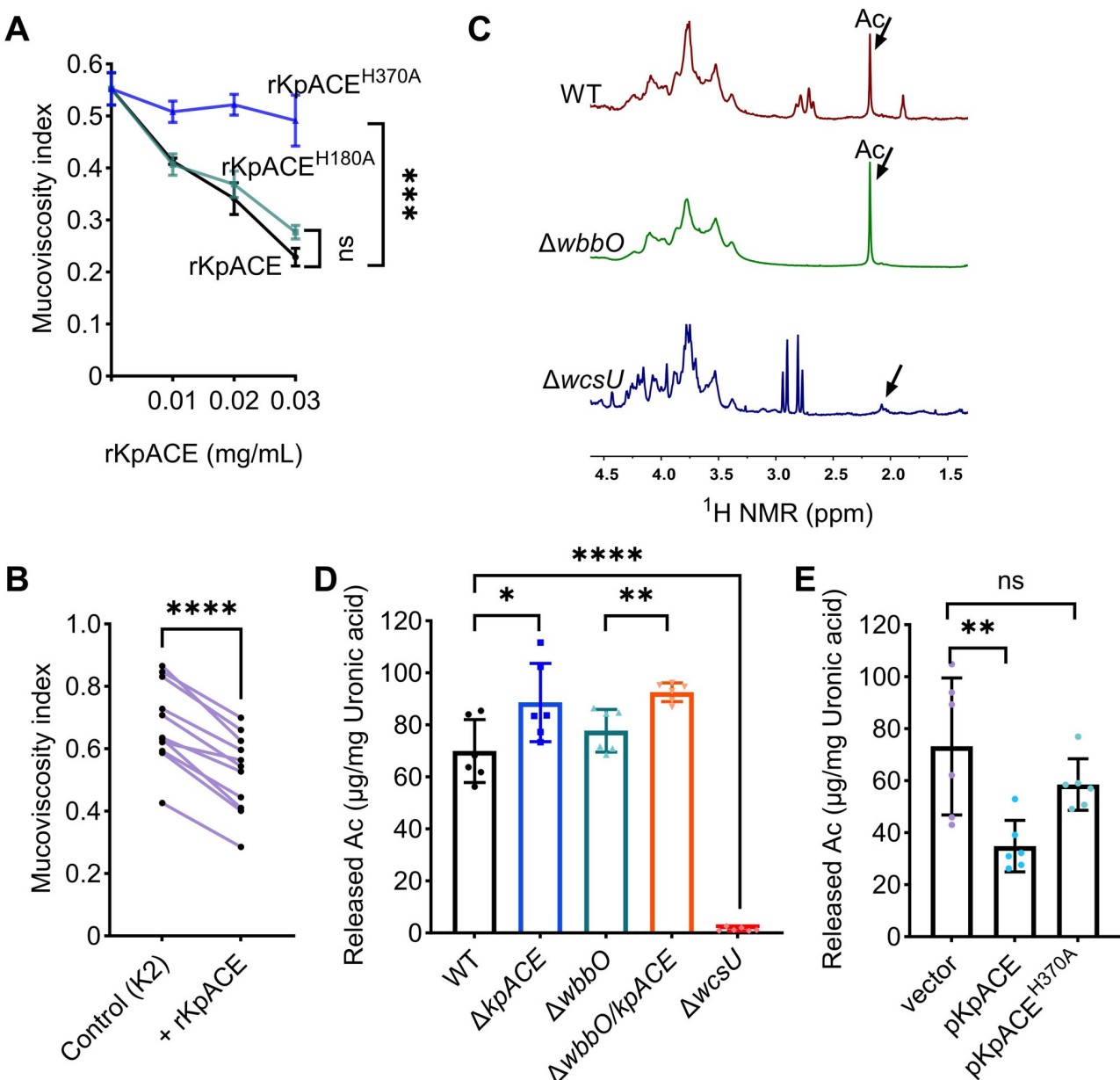

**Fig 3. KpACE acts on CPS and its deficiency increases acetylation levels in CPS. (A and B)** Mucoviscosity of ATCC43816. The *K. pneumoniae* cultures were grown in LB broth supplemented with different concentrations of the recombinant KpACE protein. **(A)** Mucoviscosity of ATCC43816 under different concentrations of either rKpACE, rKpACE^H180A, or inactive format rKpACE^H370A. The unpaired *t*-test was performed to determine differences when treated with 0.03mg/mL of rKpACE. **(B)** Mucoviscosity of eleven clinical *K. pneumoniae* strains of K2 serotype (S3 Table) in the absence or presence of 0.06 mg/ml of rKpACE. The paired t-test was performed to determine statistically significant differences. **(C)** ^1H NMR chemical shift (ppm) of EPS from wild-type (WT), Δ*wbbO*, and Δ*wcsU* in D$_2$O at 298 K. ^1H position of approximately 2.1 ppm corresponds to an acetyl group (Ac, black arrow). The spectra were representative of three independent experiments. **(D-E)** Quantification of the NaOH-released acetic acids (Ac) from exopolysaccharides (EPS) preparations. The amount of Ac was normalized by the uronic acid content of each EPS. The released Ac from EPS of kpACE-knockout mutant (D) and kpACE overexpression isogenic strains (E) were determined, respectively. The Δ*wcsU* mutant was used as a negative control. Data is presented as mean ± SD from six biological replicates. The unpaired t-test was performed to determine statistically significant differences.

effect of the rKpACE protein on eleven additional clinical strains of K2 serotype, which had been previously determined as HMV and HvKP strains in a murine bacteremia model [25]. The rKpACE protein is functional when added to cultures of these eleven clinical K2 isolates, reducing their mucoviscosity (Fig 3B). These observations suggested that KpACE acts selectively on extracellular components of K2 serotype strains as a carbohydrate acetylesterase. To further test it, we analyzed the chemical structures of EPS and its interaction with KpACE. First, we analyzed monosaccharide compositions using high-performance liquid chromatography (HPLC). The monosaccharide components of the wild-type EPS consisted of glucose, mannose, glucuronic acid, and galactose, with a molar ratio of approximately 2:1:1:1 (S8 Fig). The CPS-K2 of *K. pneumoniae* comprises glucose, mannose, and glucuronic acid, with a molar ratio of 2:1:1 [26], while the O-antigen consists of repeat units of galactose [27]. Therefore, the carbohydrate components of EPS were constituted by the combination of CPS and O-antigen. To further distinguish the contribution of EPS components, we constructed a Δ*wbbO* mutant (TH17000), lacking O-antigen (S8 Fig), to separate the CPS from the O-antigen.

Subsequently, we examined whether acetyl modifications exist in the EPS by a proton nuclear magnetic resonance spectroscopy ($^1$H NMR) analysis. The $^1$H NMR spectrum of EPS prepared from the wild-type cells displayed proton signals of acetyl group near 2.0 ppm (Fig 3C). We compared the $^1$H NMR spectrum of the Δ*wbbO* mutant to that of the wild-type and observed that the peak at near 2.0 ppm remained (Fig 3C), suggesting that the acetyl decoration is associated with CPS. Although previous NMR analysis of CPS-K2 from ATCC43816 did not detect acetylation [28, 29], a recent report has shown *O*-acetylation of CPS-K2 at *O*-6 of mannose [15], which is consistent with our findings and strongly supports the idea that KpACE deacetylates CPS-K2.

The ATCC43816 CPS biosynthesis locus harbors a gene *wcsU* (VK055_5024), which encodes a putative acyltransferase. To determine if WcsU is responsible for acetyl substitution, we constructed a deletion mutant, Δ*wcsU* (TH16622), and examined its EPS using $^1$H NMR. As expected, the inactivation of *wcsU* resulted in the loss of the acetyl peak in the $^1$H NMR spectrum of EPS (Fig 3C), indicating that WcsU is responsible for the presence of the acetyl group in CPS. Furthermore, we analyzed the presence of the *wcsU* gene in the aforementioned 11 clinical strains. Ten strains encode the full length WcsU protein, while one strain, TH13018, having a three amino acid deletion in its WcsU (S9 Fig). The WcsU homologues are highly conserved, exhibiting 99.59% identity (S9 Fig).

The activity of KpACE was then evaluated by comparing the acetylation levels of EPS in different isogenic strains, monitoring the release of acetic acids from EPS following alkali treatment as previously described [30]. The released acetic acids from both the Δ*kpACE* mutant and the Δ*wbbO*/*kpACE* double mutant (TH17108) were significantly higher than those from the wild-type and the Δ*wbbO* mutant, respectively (Fig 3D), suggesting that the KpACE deficiency led to increased acetyl level in CPS. Conversely, the released acetic acids from WTpKpACE were significantly lower than those from the control. The inactive WTpKpACE$^{H370A}$ and wild-type control released comparable levels of acetic acids (Fig 3E). The overexpression of *kpACE* led to a reduction in acetyl levels of CPS. These findings suggest that KpACE functions as a deacetylase, contributing to the deacetylation of CPS.

Our findings highlight the importance of two enzymes, acetyltransferase WcsU and acetylesterase KpACE, in the acetylation process of the capsule. WcsU adds acetyl groups to the capsules, while KpACE's catalytic activity removes acetyl groups.

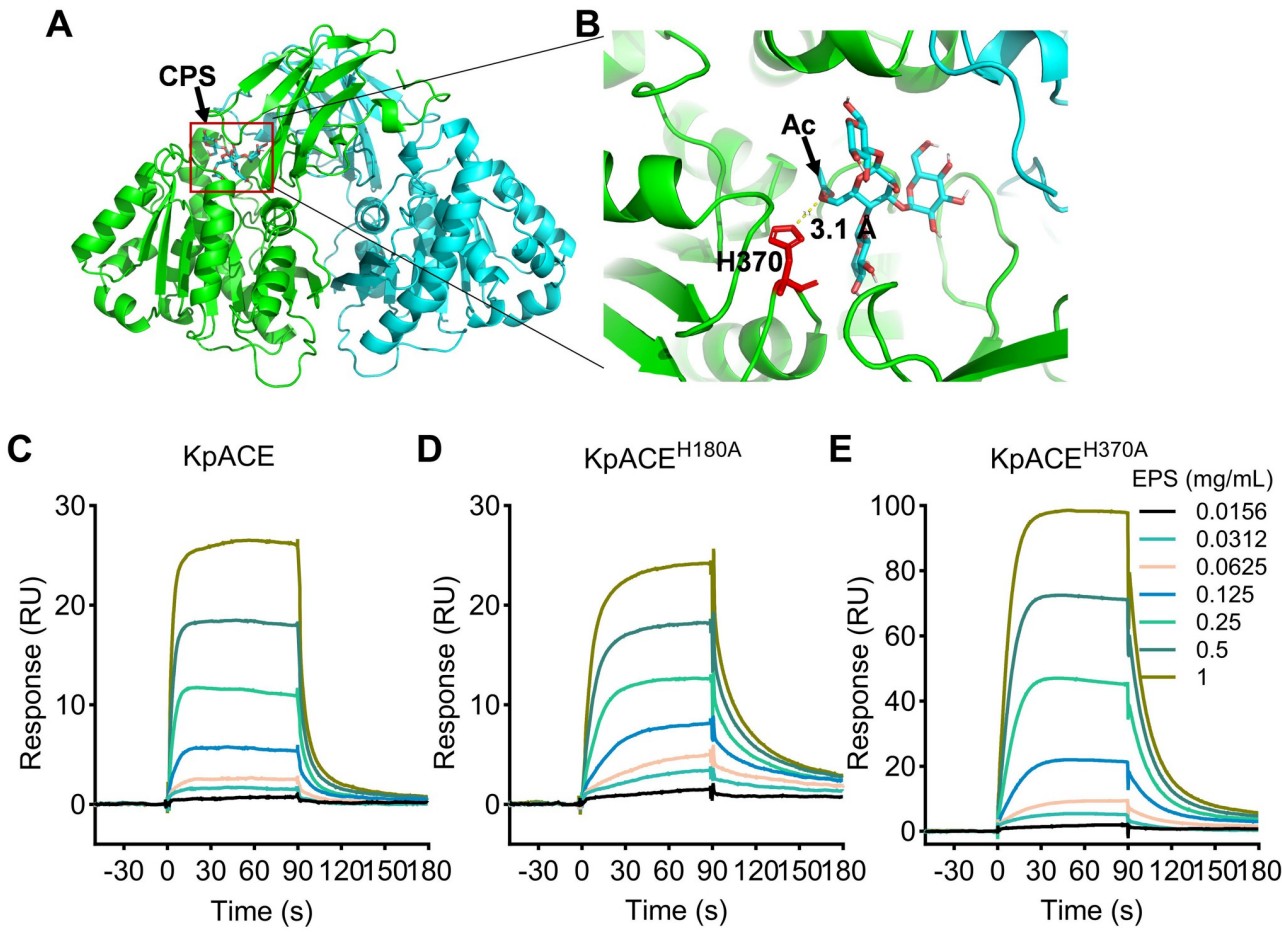

**Fig 4. Potent binding of CPS to KpACE. (A)** The best-fitted molecular docking of CPS calculated using the software AUTODOCK 4. The CPS repeat unit was highlighted as CPS (black arrow). The binding region between KpACE and one CPS repeat unit is marked in a rectangle, which is highlighted in panel B. **(B)** The distance between the activity site H370 and acetyl-group (Ac) (black arrow) is 3.1 Å. Structure figures are visualized using PyMol. **(C)** Sensorgrams of binding in resonance units (RU) of CPS (ranging from 0.0156 to 1 mg/mL) to rKpACE, rKpACE$^{H180A}$, rKpACE$^{H370A}$ measured by surface plasmon resonance. The curves are derived from the representative of three biological replicates.

## Potent binding of CPS to KpACE

Enzymes often act on their substrates via their binding, so we employed molecular docking to predict the binding interaction between KpACE and the repeat unit of CPS, a potential ligand in this context. As the rKpACE exists in a dimeric form (S6A Fig), we first predicted the dimeric three-dimensional structure of KpACE using the AlphaFold through the Google Colab server [31] (S10A Fig). Next, we defined the binding pocket of the KpACE protein, where the ligand (CPS repeat unit) is expected to bind, using the FT map tool [32]. The three-dimensional structure of one CPS repeat unit was generated using the GlyCAM tool [15,33] (S10B Fig). Molecular docking of the CPS repeat unit to the KpACE protein was performed using the software AUTODOCK 4 [34]. The best docking pose of CPS with the KpACE protein was identified, and its binding energy was determined to be -3.96 (Fig 4A), suggesting that the ligand (CPS) binds tightly to KpACE. Analyzing the binding pocket revealed that the acetyl group of CPS is close to the catalytic site H370 of KpACE, with a distance of 3.1 Å (Fig 4B), suggesting that KpACE is likely to act on CPS in deacetylation. This conjecture was further validated experimentally using the surface plasmon resonance (SPR) analysis.

**Table 1. The kinetic parameters of CPS binding to KpACE determined by SPR.**

| Protein | Association rate constant $K_a$ ($M^{-1}$ $s^{-1}$) | Dissociation rate constant $K_d$ ($s^{-1}$) | Equilibrium dissociation constants $K_D$ (M) |
|---|---|---|---|
| rKpACE | $(5.19 \pm 0.23) \times 10^5$ | $0.15 \pm 0.01$ | $(2.87 \pm 0.10) \times 10^{-7}$ |
| rKpACE$^{H180A}$ | $(1.17 \pm 0.57) \times 10^6$ | $0.06 \pm 0.03$ | $(5.35 \pm 0.52) \times 10^{-8}$ |
| rKpACE$^{H370A}$ | $(2.79 \pm 0.04) \times 10^5$ | $0.05 \pm 0.002$ | $(1.97 \pm 0.07) \times 10^{-7}$ |

The binding between CPS and KpACE was illustrated by representative binding curves from the SPR sensorgrams (Fig 4C). The kinetic parameter values are presented in Table 1. Notably, rKpACE$^{H180A}$ exhibited the strongest binding affinity toward CPS, with a $K_D$ of $5.35 \times 10^{-8}$ M, while the inactive variant, rKpACE$^{H370A}$, exhibited the weakest binding with a $K_D$ of $1.97 \times 10^{-7}$ M. As previously determined, the esterase activity level of rKpACE$^{H180A}$ is approximately 10-fold higher than that of rKpACE. These data suggest a correlation between binding affinity and enzymatic activity. Nevertheless, our findings demonstrate the interaction between CPS and KpACE.

### *In vitro* attenuation of virulent determinants by the acetylesterase activity of KpACE

To address the importance of the catalytic activity of the KpACE protein in mediating virulence traits, we examined human serum resistance [35] and biofilm formation [36], two representative virulence factors when overexpressing either KpACE or its variants.

We conducted a serum survival assay to assess the ability of *K. pneumoniae* to survive in the blood. The overexpression of either pKpACE or pKpACE$^{H180A}$ increased sensitivity to serum killing compared to the wild-type control, whereas the inactive variant pKpACE$^{H370A}$ did not increase serum killing (Fig 5A). Serum resistance is often used to evaluate the contribution of the capsule to virulence [37]. The pKpACE$^{H370A}$ mutation abrogated its catalytic activity and reversed the increased susceptibility to serum killing observed in the pKpACE overexpressing strain. These data suggest that the esterase activity of the KpACE protein plays a crucial role and exerts its effect on CPS, which influences the bacteria's sensitivity to serum killing.

The overexpression of either pKpACE (TH16356) or its active variant, pKpACE$^{H180A}$ (TH16539), decreased biofilm formation significantly, implying that the catalytic activity of KpACE is crucial for biofilm formation. The active enzyme may hydrolyze the biofilm carbohydrate matrix, thereby reducing biofilms. The inactive variant, pKpACE$^{H370A}$ (TH16541), increased biofilm formation compared to the wild-type control (Fig 5B). The substantial increase in biofilm formation observed with the dead enzyme pKpACE$^{H370A}$ likely results from the binding of inactive KpACE$^{H370A}$ to the biofilm polysaccharide without degrading the matrix. In support of this notion, we observed that an isogenic strain overexpressing the CBM48 domain alone (pKpACE$^{CBM}$, TH16538) also dramatically increased biofilm formation (Fig 5B). In contrast, the *K. pneumoniae* variant overexpressing the enzyme catalytic domain alone (pKpACE$^{Cat}$, TH17155) decreased the biofilm formation. Together, these findings suggest that the catalytic activity of KpACE is essential for its role in the degradation of biofilm matrix, while the binding of KpACE to the carbohydrate matrix significantly enhances biofilm formation.

### WcsU-deficiency reduces mucoviscosity and virulence

To further explore if CPS acetylation mediates HMV formation, we employed an acetylation-deficient mutant Δ*wcsU* to study its influence on mucoviscosity. As expected, a significant

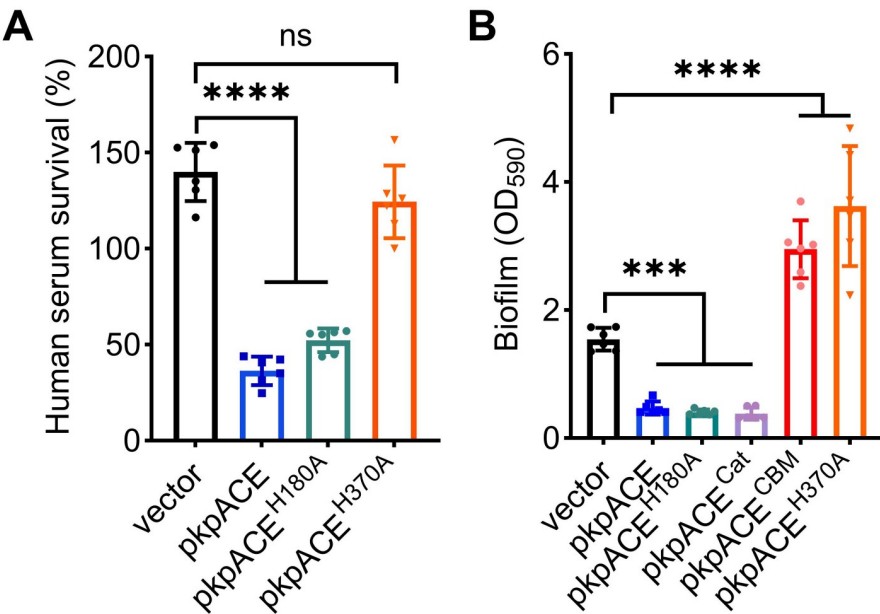

**Fig 5. Effects of kpACE overexpression on *in vitro* virulent determinants. (A)** Serum killing of *K. pneumoniae* overexpressing different KpACE variants. Serum survival percentages were calculated by dividing the starting CFUs by those enumerated after the 1-h incubation in human serum. **(B)** Biofilm formation by *K. pneumoniae*. The 24-h biofilms, formed by *K. pneumoniae* carrying either an empty vector, pkpACE, pkpACE$^{H180A}$, pkpACE$^{Cat}$, pkpACE$^{CBM}$, or pkpACE$^{H370A}$, were determined by crystal violet staining at OD$_{590nm}$. Data are presented as mean ± SD from six biological replicates. One-way ANOVA with Dunnett's multiple comparisons test was performed to determine the statistical significance.

reduction in mucoviscosity was seen in the Δ*wcsU* mutant, suggesting the importance of acetyl decoration on CPS (Fig 6A). This observation is consistent with the notion that KpACE over-expression leads to a decrease in mucoviscosity by deacetylating CPS.

Effects of the *wcsU* gene on serum killing sensitivity and biofilm formation were also investigated, revealing that the Δ*wcsU* mutant exhibited decreased serum resistance compared to the wild-type strain (Fig 6B). Conversely, the deletion of the *wcsU* gene enhanced biofilm formation (Fig 6C). Furthermore, we investigated whether the WcsU deficiency influences *in vivo* virulence using a murine pneumonia model. The Δ*wcsU* mutant showed higher survival rates than the wild-type counterpart (Fig 6D). Although the bacteria colonization in the primary lung lesion was comparable to the wild-type counterpart (approximately $10^8$ CFU/g) (Fig 6E), Δ*wcsU* exhibited lower bacterial burdens in the metastatic sites of the liver (approximately $10^5$ CFU/g) (Fig 6F) and spleen (approximately $10^5$ CFU/g) (Fig 6G). The reduction in virulence is evident when mice were infected with the *K. pneumoniae* variant lacking CPS acetylation, suggesting the role of CPS acetylation in virulence.

## Discussion

Highly virulent *K. pneumoniae* is an emerging life-threatening pathogen that poses significant public health challenges. HMV is the hallmark of its virulence, with CPS being its key determinant [38]. The degree and position of O-acetylation within CPS are critical in influencing the biological properties of some pathogenic bacteria [13,39]. Although the CPS in *K. pneumoniae* is frequently modified with acetyl groups [5–7], the impact of CPS acetylation on HMV and pathogenicity, as well as the underlying molecular mechanisms, is not well understood.

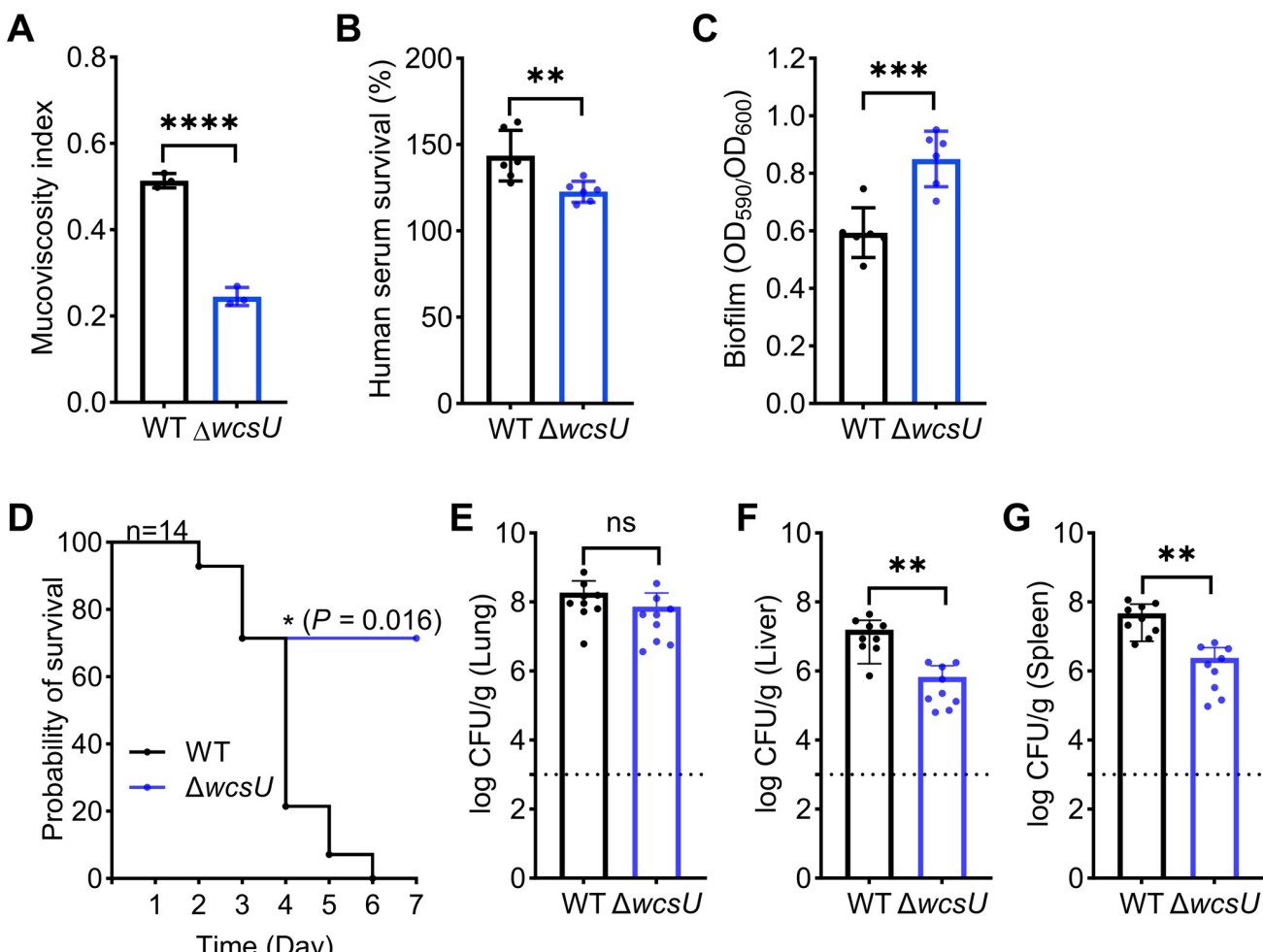

**Fig 6. Effects of CPS deacetylation on mucoviscosity and virulence. (A)** Mucoviscosity. Mucoviscosity of the Δ*wcsU* mutant was compared to that of the wild-type strain. Data are presented as mean ± SD from three biological replicates. The unpaired t-test was performed to determine statistically significant differences between the two groups. **(B)** Serum killing of *K. pneumoniae* variants. Serum survival percentages were calculated by dividing the starting CFUs by those enumerated after the 1-h incubation in human serum. **(C)** Biofilm formation by *K. pneumoniae*. The 24-h biofilms, formed by *K. pneumoniae* variants, were determined by crystal violet staining at OD$_{590nm}$. Data are presented as mean ± SD from six biological replicates. **(D-G)** Effects of Δ*wcsU* mutant on virulence in a pneumonia model using both female and male mice. (D) Survival rates of mice (10 female and 4 male) infected by *K. pneumoniae* variants. Mouse survival rates were monitored 7 days post intranasal infection with 2,000 CFU of various ATCC43816 strains. The Gehan-Breslow-Wilcoxon test was performed to compare the survival rates. Bacterial burdens in lungs (E), livers (F), and spleens (G) of infected mice (6 female and 3 male) were determined 48 h post intranasal inoculation. Each dot represents one mouse, and dotted lines represent the detection limit. An unpaired *t*-test was performed to determine the statistical significance. The separate results for female and male mice were presented in S12 Fig.

In this study, we identified that a putative acetyltransferase, WcsU, and a new acetylesterase, KpACE, form a regulatory pair for acetylation/deacetylation, modulating the *O*-acetylation level of CPS. This dynamic regulation is critical for controlling HMV formation and pathogenesis of K2 serotype *K. pneumoniae*. We found both reduction of acetylation by overexpression of *kpACE* and elimination of acetylation by inactivation of *wcsU* led to decreased HMV *in vitro* and reduced virulence *in vivo*.

Our findings align with previous studies that have shown an association between bacteria's ability to regulate CPS modification through intracellular *O*-acetylation mechanisms and their overall fitness [40, 41]. However, it has been reported an HMV-positive K2 strain Kp52.145 lacks CPS acetylation due to the absence of an intact *wcsU* gene [15,26]. In our study, the

Δ*wcsU* mutant of the ATCC43816 strain, also lacks CPS acetylation similar to Kp52.145, but reduces mucoviscosity. It is worth noting that the ATCC43816 Δ*wcsU* did not completely eliminate HMV. This suggests that the CPS acetylation may only partially contribute to the HMV formation, and the cellular context of different strains may mediate HMV differently. These studies also imply that WcsU and KpACE may differentially regulate HMV-independent virulence. In fact, the Δ*wcsU* mutant exhibited enhanced biofilm formation, contrasting with the reduction in biofilm formation observed upon overexpression of KpACE. The KpACE protein sequence of Kp52.145 shares 99.49% identity with that of the ATCC43816 strain, indicating that a functional KpACE is likely expressed in Kp52.145. These studies highlight that KpACE is multifunctional and may have additional activities beyond deacetylation that contributes to the reduction in HMV formation and virulence.

We have determined KpACE as an active acetylesterase through biochemical and genetic studies. KpACE is capable of removing acetyl groups from CPS. The KpACE protein is a multidomain esterase and is closely related to DmCE1B_ct (PDB:7b6b) from a gut Bacteroidetes *D. mossii* [42]. DmCE1B, a component of the polysaccharide utilization locus PUL17, is known for its role in starch hydrolysis [24]. The structural similarity led us to hypothesize that KpACE also hydrolyzes carbohydrates, potentially altering the structure or function of CPS, a critical virulence factor in *K. pneumoniae*. Interestingly, the strain pKpACE overexpressing KpACE, despite having a higher level of CPS, exhibited reduced virulence. This unexpected finding suggests that the modulation of CPS by acetylation, likely through the action of KpACE, plays a significant role in the virulence of *K. pneumoniae*. While the role of the enhanced KpACE activity in HMV and virulence is clearly defined and significant, the role of the KpACE deficiency is more complex. The *kpACE* mutant did not significantly alter *K. pneumoniae* virulence. It is possible that other unknown KpACE-like esterases exist in the genome that can compensate the lack of KpACE, or finetuned regulation of KpACE is essential for its manifestation in reducing HMV and virulence. Another possibility is that the animal model used in this study did not have high resolution to resolve the subtle difference between the wild type *K. pneumoniae* and its *KpACE* mutant. Further investigation on the mechanisms how KpACE or KpACE-like enzymes modulate acetylation and virulence would provide new insights into this complex and dynamic interplay between CPS acetylation and virulence.

Our findings revealed that rKpACE dose-dependently reduced mucoviscosity in various K2 serotype strains *K. pneumoniae*, likely acting selectively on the CPS component. The identification of acetyl groups in CPS of the ATCC43816, a K2 strain, supports this conjecture. Additionally, the presence of *wcsU* gene in genomes of many clinical strains implies a potential for CPS acetylation status, suggesting that the reduction in mucoviscosity following treatment with rKpACE is due to deacetylation. Although direct evidence of KpACE cleaving the acetyl group from CPS is lacking, the biochemical analysis demonstrated its deacetylation activity, supported by the release of acetic acids from the EPS of the Δ*kpACE* mutant and CPS of the Δ*wbbO/kpACE* double mutant. Molecular docking analysis and SPR studies further validated the strong binding of CPS to rKpACE protein, suggesting its involvement in enzymatic activity. Consistent with the activity seen with the exogenous addition of rKpACE, we observed that KpACE secreted into culture supernatants. Surprisingly, the extracellular form of KpACE is slightly larger than its intracellular precursor. Whether the secreted KpACE is modified and represents a more active form awaits further investigation. In the *K. pneumoniae* SK1-CPS, the native polymer chain creates a highly viscous and liquid-like environment around the cells, possibly due to the irregular acetyl substitution pattern. The removal of acetyl groups from SK1-CPS promotes the aggregation of CPS [8]. The reduction of acetylation by KpACE inhibited HMV formation, likely due to the altered secondary structure of the CPS-K2. Therefore,

more direct structural evidence is needed to define the function of KpACE in the deacetylation of the CPS.

The study also highlighted the KpACE's deacetylation activity directly contributes to serum-resistant alternations, showing that strains overexpressing pKpACE or its active variant pKpACE^H180A exhibited diminished serum resistance, while the catalytically inactive variant pKpACE^H370A did not affect serum resistance. Additionally, the Δ*wcsU* mutant exhibited decreased serum resistance, implying that the *O*-acetylation modification on CPS plays a role in shielding the bacteria from serum killing. These observations align with the hypothesis that deacetylation of CPS by KpACE may change the capsule's secondary structure, increasing susceptibility to serum. Interestingly, previous research indicated that CPS acetylation reduced the serum resistance of a K57 variant of *K. pneumoniae* [43], highlighting the complex role of CPS modifications in bacterial defense mechanisms. The specific degree and positioning of *O*-acetylation within the capsule critically influence the biological properties of pathogenic bacteria [39]. Notably, KpACE primarily modulates the level of acetyl modification rather than eliminating acetylation. This finetuned regulation explains the discrepancy in our results compared to studies focused on the complete loss of CPS acetylation.

Additionally, KpACE overexpression significantly reduced biofilm formation in *K. pneumoniae*. Considering the critical role of exopolysaccharides, including bacterial cellulose and poly-N-acetylglucosamine, in forming biofilms [44], it is plausible that KpACE's esterase activity contributes to the cleavage of ester linkage within its polysaccharide biofilm matrix, thereby diminishing biofilms. Intriguingly, strains expressing inactive variants of KpACE, WTpKpACE^H370A and WTpKpACE^CBM, exhibited enhanced biofilm formation. This suggests the involvement of the CBM48 domain in promoting biofilm formation through its interactions with the carbohydrate matrix. In line with this, the CBM48 has been shown to bind to carbohydrates [45]. Conversely, expressing only the catalytic domain without the CBM48 significantly reduced biofilm formation, indicating that the CBM48 domain is not essential for KpACE's esterase activity in degrading the biofilm matrix. This contrasts with the recognized importance of the CBM48 for the activity of related esterases such as the feruloyl esterase [45] and the 1,4-α-glucan branching enzyme [46], suggesting unique biochemical characteristics of the CBM48 domain of KpACE. These findings indicate a dual functionality of KpACE's CBM48 and catalytic domains, influencing biofilm dynamics differentially. This dual functionality necessitates further experimental exploration to fully understand the implications of KpACE's structure on its enzymatic and biofilm-regulating activities. Additionally, the elimination of *O*-acetylation from the CPS by the Δ*wcsU* mutant enhanced biofilm formation. Hence, the correlation between the level of *O*-acetylation on CPS and the process of biofilm formation warrants further in-depth research to better understand the complex and dynamic mechanisms.

Overall, our study demonstrated KpACE is active towards serotype K2 *K. pneumoniae*. However, it is unknow if K2 strain KpACEs are functional in other serotypes of *K. pneumoniae*. The prevalence of KpACE in 82.9% of *K. pneumoniae* genomes and its frequent distribution in various serotypes, KL64, KL47, KL2, and KL1, underscore its importance and the need for further research into its broader biological significance. Future research is required to systematically investigate the role of KpACE in other serotypes of *K. pneumoniae*, utilizing both *in vitro* and *in vivo* models, which would elucidate broader biological significance of KpACE homologues. Of particular interest is the potential therapeutic application of purified KpACE in treating *Klebsiella* infections, which merits additional research.

In summary, this study elucidates the vital function of WcsU and KpACE in regulating CPS acetylation and deacetylation in K2 serotype *K. pneumoniae*, affecting HMV formation, biofilm formation, serum resistance, and overall pathogenesis. These insights into CPS acetylation

dynamics offer new avenues for therapeutic strategies against *K. pneumoniae* infection, highlighting the potential for targeting these molecular processes to mitigate bacterial virulence.

## Materials and methods

### Ethics statement

This study was approved by the Institutional Animal Care and Use Committee (IACUC) of Laboratory Animal Resources Center, Tsinghua University (No. THU-02-2023-0038A).

### Bacterial strains and growth conditions

*K. pneumoniae* and derivative strains used are listed in S3 Table. The following antibiotics or chemicals were added when necessary: apramycin (30 μg/mL), spectinomycin (300 μg/mL), L-arabinose (0.2%), and sucrose (5%). Furthermore, we added varying concentrations of the rKpACE protein to LB broth to evaluate its impact on the mucoviscosity of different bacterial strains.

### Biochemical reagents

Acetylated monosaccharides used in this section, 1,2,3,4,6-β-D-glucose pentaacetate (CAS 604-69-3), 1,2,3,4,6-Penta-*O*-acetyl-β-D-galactopyranose (CAS 4163-60-4), and 1,2,3,4-Tetra-*O*-acetyl-β-D-xylopyranose (CAS 4049-33-6), were purchased from Beijing Chemsynlab Co., Ltd. These acetylated sugars were dissolved in DMSO and stored at -20˚C for later use.

Acetyl substrate, 7-Acetoxy-4-methylcoumarin (4-MU-Ac, CAS 2747-05-9), and its control, 4-Methylumbelliferone (4-MU, CAS 90-33-5), were purchased from Shanghai Aladdin Bio-Chem Technology Co., LTD. These reagents were dissolved in DMSO and stored at -20˚C, protected from light.

### Transcriptomic analysis

Distinct genes may be involved in the formation of HMV phenotype in cells at the log-phase and stationary phases, as suggested by previous research on the function of the OmpR [47]. Here, total RNA was isolated from log-phase bacterial cells ($OD_{600nm}$ = 0.5) grown in LB medium at 37˚C, 200 rpm. The total RNA was prepared following the procedure specified by the RNAprep pure Cell/Bacteria Kit (Tiangen Biotech, Beijing). RNA-seq of RNA samples prepared from wild-type and Δ*ompR* mutant cells was conducted at the Novogene Bioinformatics Technology (Beijing, China). The bioinformatics analyses were performed using the free online platform of Majorbio Cloud Platform (www.majorbio.com). The DEGs from the wild-type and Δ*ompR* mutant cells were identified based on a fold change ≥ 2 and a *P* value < 0.05. Subsequently, GO enrichment analysis of the DEGs was carried out to determine statistically significant differences.

### Quantitative RT-PCR (qRT-PCR)

Total RNA was prepared according to above described. cDNA was synthesized using the ReverTra Ace qPCR RT Master Mix with gDNA Remover (TOYOBO Biotech, China). Quantitative RT-PCR was performed using UltraSYBR Mixture (CWBIO, China) on a BioRad CFX96 system. Gene expression was normalized to the abundance of the 16S rRNA transcript. Relative differences in mRNA levels were calculated using the $2^{-\Delta\Delta Ct}$ method. The experiment was performed twice with two independent samples and three technical replicates for each

sample. The *galF* gene was used as a negative control [16]. Primers for qRT-PCR are listed in S4 Table.

## Construction of in-frame deletion mutants

Marker-less deletion mutants were generated by a CRISPR-Cas9-mediated genome-editing method as previously described [48]. The primers used in CRISPR-Cas9 genome editing are listed in S4 Table. Additionally, S5 Table contains details about the construction of pSGKP-sgRNA plasmids.

## Construction of KpACE overexpression plasmid

The open reading frame of KpACE and the backbone of the vector pTH16235 were amplified with primers listed in S4 Table, respectively. The gel-purified PCR products were assembled at 50˚C for 20 minutes by a NEBuilder HiFi DNA Assembly Cloning Kit (New England BioLabs, USA). Next, the assembled products were directly transferred into *E. coli* DH5α competent cells to amplify plasmids containing the *kpACE*. The recombinant plasmid was PCR amplified with a primer pair, Pr17968 and Pr17969, from transformant colonies and sequenced to select the correct colonies. Finally, the recombinant plasmid overexpressing pkpACE (pTH16351) was electro-transformed into *K. pneumoniae* ATCC43816 competent cells to generate corresponding overexpression strain WTpkpACE (TH16356). The constructions of these strains are listed in S5 Table.

## Site-directed mutagenesis of KpACE

Conserved histidine residues in the position of either 180 or 370 of KpACE were selectively changed to alanine to define the catalytic active site of KpACE. First, the plasmid pkpACE (pTH16351) was amplified with a pair of mutagenic primers with the desired mutation (S4 Table). Next, the resultant PCR product was digested by enzyme DpnI (Beyotime, Beijing) for 1 hour and then chemically transformed into *E. coli* DH5α competent cells. Subsequently, the mutated plasmids from the transformants were sequenced to confirm the presence of the desired mutations. Finally, the expected recombinant plasmid was transformed into *K. pneumoniae* competent cells by electroporation and selected by spectinomycin (300 μg/mL). The constructs of these strains are listed in S5 Table.

## Quantification of mucoviscosity and capsule

The mucoviscosity index was determined by calculating the ratio of the $OD_{600nm}$ of the supernatant to the $OD_{600nm}$ of the original culture, providing a semi-quantitative measure of the HMV phenotype [49]. Capsule quantification was performed and denoted as uronic acid (UA)/$OD_{600nm}$ as described [49].

## Electrophoretic mobility shift assay

Electrophoretic mobility shift assay (EMSA) is a useful tool for determining the binding of a transcriptional factor to its DNA binding fragments. EMSA was performed as described [50] with modifications using a Viagene Biotech Inc., USA kit.

A potential OmpR binding site, (-123)-TGTCTAACCGCAAAAATA-(-106), was predicted upstream of the VK055_3347 ATG codon, based on the online gene regulation and gene expression database PRODORIC (https://www.prodoric.de/) [51]. The appropriate promoter fragments of the VK055_3347 locus in *K. pneumoniae* ATCC43816 were amplified using the respective primer sets (S4 Table). The purified rOmpR [16] was added to the reaction mixture

at concentrations ranging from 0 to 1 μg, along with 50 ng of amplified DNA fragments, 25 mM acetyl phosphate, 1 mM MgCl$_2$, and 1 μL of poly(dI-dC). The mixture was then incubated for 20 minutes at room temperature, and the reaction was stopped by adding a loading buffer. The mixture was then separated by electrophoresis on a 6% non-denaturing acrylamide gel in 0.5× Tris Borate EDTA (TBE) buffer and stained with Gelred to visualize the bound OmpR and DNA binding complex. The promoter fragment of *gcvT* was used as a negative control [16].

### Structural prediction and similarity analysis of KpACE

KpACE is a previously undefined protein. InterPro database was used to perform functional analysis of KpACE [52]. Additionally, AlphaFold was used to predict the three-dimensional structure of KpACE [31]. The predicted protein structure was then used as a query to search against the DALI server's database (Database of Protein Alignments and Homology) of known protein structures. The DALI server identifies structurally homologous proteins [23]. Corresponding polypeptide sequences of these structurally homologous proteins were identified and stored in FASTA format. Finally, Clustal Omega was used to align multiple protein sequences to identify regions of similarity and conservation [53], and Espript 3.0 was used to visualize the alignment results [54].

### Multisequence alignment and phylogenetic analysis

A search for all complete *K. pneumoniae* genomes available on NCBI up to March 2024 yielded a total of 1,819 genomes. Subsequently, we used the ATCC43816_KpACE protein sequence as a reference for BLASTx searches, setting a cutoff for identity at 90% and an *e*-value threshold of 1e-5 [55]. Based on the comparison results, nucleotide fragments were extracted from the original sequence and translated into protein sequences using the Transeq server (https://embossgui.sourceforge.net/demo/transeq.html). MAFFT was then used for multiple sequence alignment of protein sequences, with the first amino acid specified as methionine and parameters set to achieve an identity rate of over 90% and a coverage rate exceeding 80% [56]. The alignment result was used to construct a phylogenetic tree (a maximum likelihood mode) with the IQ-TREE tool [57].

Additionally, the gene *wcsU* from clinical *K. pneumoniae* strains was amplified by PCR with the primer pair Pr19927/Pr19928 and then sequenced. Subsequently, the translated protein sequence was aligned with that of ATCC43816. DNAMAN 9.0 was used to align multiple protein sequences to identify regions of similarity and conservation and to visualize the alignment result.

### Purification of recombinant KpACE

The *kpACE* and its variants, without the signal peptide-encoding region, were amplified by PCR using primers listed in S4 Table, respectively. The purified PCR products were cloned into pET-28a using a NEBuilder HiFi DNA Assembly Cloning Kit. S5 Table contains details about constructing heterologous protein expression plasmids. The ligated products were transformed into *E. coli* BL21-DE3 cells and transformants were selected on LB agar plates supplemented with kanamycin (50 μg/mL). Each recombinant vector encodes a protein KpACE with a 6×His tag at the N-terminus. The recombinant proteins were induced by adding 0.5 mM of isopropyl β-D-1-thiogalactopyranoside (IPTG) at 28˚C for 4 hours. Cells were harvested and resuspended with Binding Buffer (20 mM Tris-HCl, 300 mM NaCl, 20 mM imidazole, pH 8.0). Next, cells were lysed by sonication on ice and centrifuged at 12,000 rpm for 30 minutes at 4˚C. The clarified supernatant was then subjected to Ni-IDA SefinoseTMResin (BBI, Shanghai). The purity of rKpACE was analyzed by 12% SDS-PAGE, and the concentration was quantified using the BCA method.

## Catalytic activity determination

To determine the acetylesterase activity of KpACE, we used a model substrate, 4-methylumbelliferyl acetate (4-MU-Ac), as described previously [58]. Optimal pH value and enzymatic kinetic parameters were determined. The data was fitted to the Michaelis-Menten equation to define the kinetic parameters, including the maximum reaction rate ($V_{max}$) and the Michaelis-Menten constant ($K_m$). Meanwhile, the enzymatic efficiency parameter of Kcat/Km was calculated.

To determine if the acetylesterase KpACE specifically targets and catalyzes the hydrolysis of acetyl ester bonds from acetylated monosaccharides substrates, 20 μg of the purified rKpACE was incubated with 2 mM of 1,2,3,4,6-β-D-glucose pentaacetate, 1,2,3,4,6-Penta-*O*-acetyl-β-D-galactopyranose, and 1,2,3,4-Tetra-*O*-acetyl-β-D-xylopyranose, respectively, in the 150 μL of reaction buffer (100 mM sodium phosphate pH 7.4) at 37°C for 1 hour. The released acetic acid was detected using the acetic acid kit (K-ACETRM 04/20, Megazyme) according to the kit's instructions and used to illustrate the enzymatic activities of KpACE.

## Western blot analysis

Overnight cultures of WTpKpACE (TH16356) were sub-cultured into 5 mL of fresh LB medium at a ratio of 1:100 until $OD_{600nm}$ reached 1.0, and 500 μL of cultures were transferred into a 1.5mL tube and centrifuged at 13,000 rpm for 10 minutes. All supernatants were then transferred to a 10 kDa Amicon Ultra-0.5 mL column (Millipore) to concentrate the volume to 80 μL, and 20 μL of 5× SDS-PAGE loading buffer was added. Additionally, the remaining cell pellets were suspended with 100 μL of 1× SDS-PAGE loading buffer. All samples were boiled for 10 minutes to lyse the cells, and 15 μL of each sample was loaded onto a 12% SDS-PAGE gel, and the KpACE-6×His expressed by TH16356 was probed using a 6×His-tag monoclonal antibody (Invitrogen). The response regulator VicR, an intracellular protein, was used as a subcellular control [59]. The image was taken using the Amersham ImageQuant 800 Western blot imaging system.

## Isolation of exopolysaccharides (EPS)

EPS was prepared from *K. pneumoniae* cells grown overnight on a BAP at 37°C, following a previously described method [60]. Briefly, cells were harvested and resuspended in PBS. Next, 1/10 volume of 1% zwittergent 3–14 in 100 mM citrate buffer (pH 2.0) was added to the cell suspension, and the mixture was incubated at 50°C for 1 hour. After centrifuging for 30 minutes at 12,000 rpm, the supernatant was filtered through a 0.22 μm filter, precipitated with 80% ethanol, and left to stand for 24 hours at -20°C. The precipitate was recovered by centrifugation and dried. For further purification [61], 600 mg of dried crude EPS was suspended in 300 mL of a solution containing 20 mM Tris-HCl (pH 8.0) and 2 mM $MgCl_2$. To this suspension, 1 mg of DNase I (Solarbio, Beijing) and 1 mg of RNase A (Takara, Beijing) were added and incubated at 37°C for 2 hours. Subsequently, 2 mg of protease K (Beyotime, Beijing) was added and incubated at 55°C for 2 hours. The resulting supernatants were then dialyzed against distilled water using a 3.5 kDa molecular weight dialysis bag for 3 days to remove insoluble impurities and lyophilized.

## Monosaccharide composition analysis of EPS by HPLC

The monosaccharide composition of *K. pneumoniae* EPS was determined and quantified using a high-performance liquid chromatography (HPLC) method combined with 1-phenyl-3-methyl-5-pyrazolone (PMP) pre-column derivatization, as described previously [62]. EPS

was disintegrated with 2 M trifluoroacetic acid (TFA) at 110°C for 6 hours and then dried to remove residual TFA. The sample or monosaccharide standard was subsequently derivatized with 0.5 M PMP in methanol at 70°C for 1 hour. 100 μL of $H_2O$ and 100 μL of 0.3 M NaOH were added to neutralize the reactions, followed by 110 μL of 0.3 M HCl. The derivatized samples were purified with chloroform three times. The analysis was performed using an Agilent Eclipse XDB-C18 column (5 μm, 4.6 × 150 mm) at a fixed flow rate of 1.0 mL/min, with a mobile phase composed of 16% $CH_3CN$ in 0.1 M PBS (pH 6.7), and the column temperature was maintained at 30°C. The fraction was monitored using an ultraviolet detector at 245 nm.

### Nuclear magnetic resonance

The EPS was initially dissolved by heating and stirring and subsequently analyzed by Nuclear Magnetic Resonance (NMR) spectroscopy [43]. Briefly, 5 mg of dry EPS were stirred with 600 μL of $D_2O$ at 60°C for 12 hours, then lyophilized. This process was repeated three times. Then, the samples were dissolved in 600 μL of $D_2O$ by stirring. The supernatant was obtained through centrifugation and loaded into the NMR tube for detection. NMR analysis was conducted using the Bruker Avance Neo 400 MHz NMR spectrometer, with $^1H$ NMR analysis performed at 298 K.

### Acetic acid assay and measurement of acetyl esters in EPS

We measured the concentration of acetic acid released from the samples using the acetic acid kit (K-ACETRM 04/20, Megazyme) according to the kit's instructions. The kit measures NADH consumption, which is detected by the decrease in absorbance at 340 nm using a microplate reader (PerkinElmer EnVision). To assess the acetyl esters in EPS, we measured the amount of released acetic acid following complete acetyl ester hydrolysis using NaOH, as described in a previous publication [30]. The amount of uronic acid in the EPS was used to normalize the acetic acid concentration.

### Molecular docking of KpACE and CPS repeat unit

Molecular docking was used to simulate the binding interactions between KpACE and the CPS-K2 repeat unit *in silico* [54]. Briefly, 3D structures of the enzyme and the polysaccharide repeat unit were prepared. The 3D structure of the ATCC4316 CPS-K2 repeat unit [15] was generated using a molecular modeling technique, GlyCAM [33]. The GlyCAM (www.glycam.org) is a server that used to generate glycan structure for modeling, docking, and simulation.

Docking of the polysaccharide repeat unit to KpACE was performed with the software AUTODOCK 4 [34]. The potential binding spots were predicted using an FT map to create a hot map [32, 63]. All structure figures were visualized using PyMol [64].

### Determination of EPS molecular weight by HPGPC-MALLS

The molecular weight of the EPS samples was estimated using high-performance gel permeation chromatography combined with a multi-angle laser light scattering (HPGPC-MALLS) analysis. The EPS sample was examined using an Agilent 1260 HPLC system (Agilent, Santa Clara, CA, USA), which was equipped with a Dawn Heleos-II multi-angle laser light scattering instrument and a refractive index detector (Wyatt Technology, Santa Barbara, CA, USA). Subsequently, gel permeation chromatography was conducted using Shodex OHpak SB-804 HQ and SB-803 HQ columns, with a flow rate of 0.6 mL/min and a mobile phase of 0.1 mol/L $NaNO_3$. The sample concentration was set at 1 mg/mL, and the analysis was conducted at 30°C for 45 minutes. The refractive index increment (dn/dc) of 0.15 mL/g was used for the

analysis [19], and the data were analyzed using the Wyatt ASTRA software (V 6.1.1). The molecular weight presents as a mean of three independent examinations.

### Surface plasmon resonance for the interaction of CPS with KpACE

The interaction between CPS and KpACE was detected by surface plasmon resonance (SPR) [65]. Since the catalytic activity of KpACE may be associated with KpACE binding to CPS, we parallelly examined the interaction between each variant, KpACE$^{H180A}$ and KpACE$^{H370A}$, and CPS, as previously described [66].

The SPR assay was conducted using a Biacore T200 SPR spectrometer (Cytiva, USA), following previously described methods [67]. A CM5 chip (GE HealthCare, USA) was first installed in the Biacore T200 to immobilize rKpACE or its variant for SPR analysis. After rinsing the chip surface with phosphate-buffered saline containing detergent (PBS-P) running buffer, the sensor chip was activated for 900 seconds with a solution containing 0.1 M NHS (N-hydroxy succinimide) and 0.4 M EDC (1-ethyl-3-[3-dimethylaminopropyl] carbodiimide) at a flow rate of 10 μL/min. Following activation, a protein solution at a concentration of 20 μg/mL was introduced into sodium acetate buffers (10 mM, pH 5.0, 5.0, and 4.5, respectively) at a flow rate of 10 μL/min. The solutions of rKpACE or variants were added to separate channels on the chip surface for 900 seconds, leaving one channel blank reference. Any unreacted activated carboxylic acid on the chip surface was blocked with 1 M ethanolamine pH 8.5 at the same flow rate for 450 seconds. Any loosely bound proteins were washed away with a mobile phase solution, resulting in final protein coupling amounts of 4737 resonance units (RU), 11444 RU, and 4589 RU in channels 2, 3, and 4, respectively.

In preparation for the CPS binding test, the baseline was allowed to stabilize for a minimum of 1 hour in a PBS-P running buffer. Subsequently, CPS (EPS-Δ*wbbO*) was dissolved in PBS-P at six different concentrations, ranging from 0.0156 to 1 mg/mL. These samples were injected for 90 seconds at a constant flow rate of 30 μL/min. Following the injection phase, a 120-second dissociation phase occurred. The response was continuously monitored at 25˚C, and the resulting data (sensorgram) was then subtracted from the response of the reference surface. We utilized BLA evaluation software version 4.1 to analyze the kinetic parameters and employed a 1:1 model for curve fitting.

### Murine infection model

A murine pneumonia model was used to evaluate *in vivo* virulence. Both male and female CD1 mice (6–8 weeks old) were intranasally inoculated with approximately 2,000 CFU of log-phase grown *K. pneumoniae*. The virulence of bacterial strains was evaluated using a survival curve of infected mice (n ≥ 6). Bacterial colonization was assessed by determining bacterial burden in multiple organs, including the lung, liver, and spleen. As no dissemination was observed at 24 hours post-infection and infected mice only started to succumb to infection 48 hours post-infection, the bacterial burden in various organs was quantified at the 48-hour time point. The data are presented as CFU/g tissue. The sample size is detailed in the figure legend.

### Biofilm formation

To determine the effects of KpACE on *K. pneumoniae* biofilm formation, we performed a crystal violet staining assay as described [68] with modifications. Biofilm biomass was quantified by measuring the absorbance at 590 nm using a microplate reader (BioTek Synergy H1). Experiments were performed with six biological replicates.

## Serum survival assay

Human serum killing of *K. pneumoniae* was used to assess *K. pneumoniae in vitro* virulence. The assay was performed as described previously with modifications [68]. Briefly, approximately $10^5$ CFU of fresh bacterial culture were prepared and added to 96-well microtiter plates in the present or absence with 20% human serum. After a 1-hour incubation at 37˚C, bacterial counts were enumerated. The survival rates were calculated by dividing the initial bacterial counts by the CFUs after 1-h incubation.

## Statistical analysis

Results were expressed as the mean ± SD. Statistical analyses were performed using GraphPad Prism v9.3.1. Significant differences are defined by *P* values of $< 0.05$ (*), $< 0.01$ (**), $< 0.001$ (***), $< 0.0001$ (****), and ns for no significance. Statistical analysis for each figure is denoted in the figure legends.

## Supporting information

**S1 Fig. Top 10 genes enriched from differentially expressed genes.** GO enrichment analysis of differentially expressed genes between the wild-type and Δ*ompR* mutant cells at the log growth phase. (A) Chord diagram of top 10 GO terms. The top 10 genes are ranked by $\log_2$FC (Fold change) on the left side of the circle, and the correlating GO terms are on the right side. The values of $\log_2$FC were presented in S1 Table. The genes annotated as carbohydrate porin are highlighted with a red rectangle. (B) The transcriptional level of select genes. The transcriptional expression of select genes was quantified using the qRT-PCR assay. The expression level of each gene in the Δ*ompR* mutant were normalized to that of wild-type (WT). The dotted line denotes the mean value of wild-type.
(PDF)

**S2 Fig. Binding of recombinant OmpR to the promoter region of VK055_3347. (A)** Diagrams of the promoter region of the VK055_3347 locus (not drawn to scale). The black box represents a putative OmpR binding sequence. Nucleotide numberings are relative to the VK055_3347 ATG codon, respectively. The relative position and length of the DNA fragments used in EMSA are shown. **(B)** EMSA results of rOmpR. The $P_{VK055\_3347}$ DNA fragment was mixed with increasing concentrations of phosphorylated rOmpR at 0, 0.3, 0.7, and 1 μg (Lanes 1–4). The $P_{gcvT}$ DNA fragment was used as a negative control, mixing with phosphorylated rOmpR at 0, 1 and 4 μg (Lanes 1–3). DNA bands were detected by Gelred staining. The positions of DNA fragments that had not shifted were labeled.
(PDF)

**S3 Fig. Effects of KpACE deficiency on virulence in a pneumonia model using both female and male mice.** (A) Survival rates of mice (6 female and 4 male) infected by *K. pneumoniae* variants. Mouse survival rates were monitored 7 days post intranasal infection with 2,000 CFU of various ATCC43816 strains. The Gehan-Breslow-Wilcoxon test was performed to compare the survival rates. (B-D) Bacterial colonization in different organs of infected mice (3 female and 3 male). Bacterial burdens in lungs (B), livers (C), and spleens (D) were determined 48 h post intranasal inoculation. Each dot represents one mouse. An unpaired *t*-test was performed to determine the statistical significance.
(PDF)

**S4 Fig. Multisequence alignment of KpACE homologues in different *K. pneumoniae* serotypes. (A)** The distribution of KL-serotypes of 1503 *K. pneumoniae* strains harboring KpACE

homologues. 1503 KpACE homologues were retrieved from the NCBI database as detailed in the Materials and Methods section. The K-serotype of each strain was determined using the Kleborate tool, which performed K-locus typing based on their genomic sequences. KL-types with a count of less than 30, combined together as 'others', were not included. (**B**) The multisequence alignment of KpACE homologues from representative *K. pneumoniae* K-serotypes. The alignment was created and visualized using DNAMAN 9.0 software. The identity of the ten KpACE homologues is 99.85%. Amino acid residues that are identical across sequences are represented in black text on a white background, while variable residues are depicted as black text on a pink background.
(PDF)

**S5 Fig. Structural prediction and similarity analysis of KpACE. (A)** Diagram of KpACE (not drawn to scale). (**B**) Three-dimensional structure of KpACE predicted by AlphaFold. Three putative classical His-Ser-Asp catalytic triads for hydrolysis, including H180-S185-D217, H291-S287-D248, and H370-S278-D339, were predicted *in silico* and denoted with rectangles. Bidirectional arrows indicate the domains for CBM48 and the Catalytic domain, respectively. (**C**) Similarity of KpACE (Cyan) and DmCE1B (PDB:7b6b) (Red). (**D**) The conserved residue H180. (**E**) The conserved H370-S278-D339 of KpACE (Cyan) with the catalytic triad (H638-S542-E606) of DmCE1B (Red). (**F**) Multisequence alignment of KpACE and its four homologues. Secondary structural elements of the closest homologue DmCE1B (PDB: 7b6b), a carbohydrate esterase from *Dysgonomonas mossii*, are shown above the alignment. Additional protein sequences are obtained from acetylesterase/feruloylesterase axe1-6A (AXFA_PRER2) from *Prevotella ruminicola*, acetyl xylan esterase (Bil1033-CE1) from *Bacteroides intestinalis*, and feruloyl esterase (Bil1039-CE1) from *Bacteroides intestinalis*. The catalytic triad (H638-S542-E606) of DmCE1B and conserved residues of Histidine of KpACE are indicated by green and black arrows, respectively. The alignment was created in Clustal Omega and visualized using Esprit 3.0. Conserved amino acid residues are displayed as red text on a white background with a blue border. Identical amino acid residues between sequences are displayed in white text on a red background with a blue border.
(PDF)

**S6 Fig. Kinetic activity of recombinant KpACE. (A)** Size-exclusion chromatography purification of recombinant KpACE. Purification of the rKpACE protein using molecular size Superdex 200. Samples corresponding to the peak position at 14.38 mL (75.6 kDa) were collected, and the presence of the KpACE protein was confirmed by 12% SDS-PAGE gel electrophoresis followed by protein staining. (**B**) Kinetic acetylesterase activity of rKpACE determined with 4-MU-Ac at pH 7.4. The data were nonlinearly fitted to the Michaelis-Menten equation by GraphPad Prism v9.3.1. (**C**) Hydrolytic activities of KpACE on three fully acetylated monosaccharides. Acetylated monosaccharides include 1,2,3,4-Tetra-O-acetyl-β-D-xylopyranose (Xyl), 1,2,3,4,6-Penta-O-acetyl-β-D-galactopyranose (Gal), 1,2,3,4,6-β-D-glucose pentaacetate (Glu). The activity was determined by measuring the amounts of acetic acids released from the substrates by rKpACE and its variants. Data are from six biological replicates. The unpaired *t-test* was performed to determine significant differences.
(PDF)

**S7 Fig. Extracellular localization of the KpACE protein.** Immunoblotting of culture supernatants and cell pellets was performed on *K. pneumoniae* cells carrying pkpACE (TH16356), which were cultured in an LB medium to $OD_{600nm}$ of 1.0. All samples were boiled for 10 minutes, and 15 μL of each sample was loaded onto a 12% SDS-PAGE gel. The KpACE-6×His fusion protein expressed by TH16356 was probed using a 6×His Tag monoclonal antibody

(Invitrogen). The response regulator VicR, a known intracellular protein, was employed as a negative control to verify the localization of KpACE. Comparing the immunoblotting results of VicR and the KpACE-6×His fusion protein, we conclude that any detected signal for KpACE in the supernatant was not due to cell lysis or contamination with intracellular proteins.
(PDF)

**S8 Fig. The monosaccharide compositions of EPS. (A)** Monosaccharide compositions of wild-type EPS determined using High-Performance Liquid Chromatography (HPLC). **(B)** The molar ratio of monosaccharides in EPS from wild-type and *ΔwbbO* strains.
(PDF)

**S9 Fig. Multisequence alignment of WcsU protein from 11 clinical strains of *K. pneumoniae*.** The gene *wcsU* from 11 clinical strains was amplified and sequenced. Subsequently, the translated protein sequence was aligned with that of ATCC43816. The alignment was created and visualized with DNAMAN 9.0 software. Identical amino acid residues between sequences are displayed in black text on a white background, while variable residues are depicted as black text on a pink or cyan background. The mean identity is 99.59%.
(PDF)

**S10 Fig. The predicted three-dimensional structures of KpACE and CPS repeat unit. (A)** Three-dimensional (3-D) structure of KpACE was predicted using AlphaFold through the Google Colab server. **(B)** The 3-D structure of one CPS repeat unit was generated by GlyCAM based on the CPS repeat unit of four sugars (upper), and the acetyl group was marked as Ac (bottom) (see Materials and methods).
(PDF)

**S11 Fig. *kpACE* overexpression reduces virulence in a pneumonia model using both female and male mice. (A-D)** *kpACE* overexpression (pkpACE) reduces virulence in a pneumonia model using female mice. (A) Survival rates of mice. Bacterial burdens in lungs (B), livers (C), and spleens (D) were indicated, respectively. **(E-H)** *kpACE* overexpression (pkpACE) reduces virulence in a pneumonia model using male mice. (E) Mouse survival rates of mice. Bacterial burdens in lungs (F), livers (G), and spleens (H) were indicated respectively. Mouse survival rates were monitored 7 days post intranasal infection with 2,000 CFU of ATCC43816 variants. Bacterial burdens in organs were determined 48 h post intranasal inoculation. Each dot represents one mouse, and dotted lines represent the detection limit. The Gehan-Breslow-Wilcoxon test was performed to compare the survival rates. One-way ANOVA with Dunnett's multiple comparisons test was performed to determine the statistical significance among bacterial burdens in organs.
(PDF)

**S12 Fig. Effects of CPS deacetylation on virulence in a pneumonia model using both female and male mice.** (A-D) CPS deacetylation (ΔwcsU) reduces virulence in a pneumonia model using female mice. (A) Survival rates of mice. Bacterial burdens in lungs (B), livers (C), and spleens (D) were indicated, respectively. (E-H) CPS deacetylation (ΔwcsU) reduces virulence in a pneumonia model using male mice. (E) Survival rates of mice. Bacterial burdens in lungs (F), livers (G), and spleens (H) were indicated respectively. Mouse survival rates were monitored 7 days post intranasal infection with 2,000 CFU of various ATCC43816 strains. Bacterial burdens in organs were determined 48 h post intranasal inoculation. Each dot represents one mouse. The Log-rank (Mantel-Cox) test was performed to compare the survival rates. Unpaired t-test was performed to determine the statistical significance among bacterial

burdens in organs.
(PDF)

**S1 Table. Differentially expressed genes in planktonic cells of the Δ*ompR* mutant.**
(DOCX)

**S2 Table. KL-serotype of each strain among 1503 *K. pneumoniae* harboring KpACE homologues (related to S4A Fig).**
(XLSX)

**S3 Table. Characteristics of strains in this study.**
(DOCX)

**S4 Table. Primers used in this study.**
(DOCX)

**S5 Table. Characteristics of isogenic strains constructed in this study.**
(DOCX)

**S6 Table. Raw data set for figures in this study.**
(XLSX)

**S1 Data. Data include whole genome sequencing of 4 isogenic mutants in this study, alignment of KpACE homologues among 1503 genomes for S4 Fig, and phylogenetic tree of KpACE among 1503 genomes for S4 Fig.**
(ZIP)

## Author Contributions

**Conceptualization:** Chao Cai, Hui Wu.

**Data curation:** Jun Guo, Chao Cai.

**Formal analysis:** Zhe Wang, Hua Zhang, Shuaihua Fan.

**Funding acquisition:** Lijun Wang, Jing-Ren Zhang.

**Investigation:** Lijun Wang, Zhe Wang, Hua Zhang, Qian Jin.

**Methodology:** Yanni Liu, Xueting Huang.

**Supervision:** Jing-Ren Zhang, Hui Wu.

**Writing – original draft:** Lijun Wang.

**Writing – review & editing:** Hui Wu.

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
