## [Decision Letter · Decision Letter 0]

28 May 2024

Dear Professor Wu,

Thank you very much for submitting your manuscript "A novel esterase regulates Klebsiella pneumoniae hypermucoviscosity and virulence" for consideration at PLOS Pathogens. Your manuscript was reviewed by members of the editorial board and by three independent reviewers. Although there was some difference of opinion among the reviews (below), we would like to invite the resubmission of a significantly revised version that addresses the reviewers' comments.

There are several key issues must be addressed. A major concern raised by both Reviewers 1 and 2 regards the extension of the results to other strains and other capsular serotypes: are these enzymes only important for serotype K2? A phylogenetic analysis would be helpful to put the findings in context. Also as noted by Reviewers 1 and 2, the strains and constructs should be comprehensively tested in the various described assays. Furthermore, as indicated by Reviewer 3, how did you determine the structure of the K2 capsule described in Figure S8 B? What is GlyCAM? If this is predicted, then the algorithm used and the basis for the model should be indicated; if this structure is from a published report (such as Reference 15), then this should be cited. This is important since prior studies suggest that K2 capsule is hyperviscous, but lacks acetylation. Lastly, all three reviewers had concerns about missing or inadequate controls that need to be experimentally addressed.

There are also issues regarding the description of results and conclusions: caution must be used by being specific about the strain or adding appropriate contextualization to help readers understand the extent that the findings are applicable.

We cannot make any decision about publication until we have seen the revised manuscript and your response to the reviewers' comments. Your revised manuscript is also likely to be sent to reviewers for further evaluation.

Sincerely,

Vincent T. Lee

Academic Editor

PLOS Pathogens

D. Scott Samuels

Section Editor

PLOS Pathogens

Michael Malim

Editor-in-Chief

PLOS Pathogens

orcid.org/0000-0002-7699-2064

Reviewer's Responses to Questions

**Part I - Summary**

Reviewer #1: The strength of this study is the topic of hypermucoviscosity in Klebsiella pneumonia and finding genes encoding enzymes that modify the structure.

Weaknesses include the lack of clarity was to whether these findings are applicable to strains of other capsular types (are these enzymes only important for serotype K2?) as well as the over interpretation and lack of controls in some of some of the results and experiments.

Reviewer #2: This manuscript reports the findings of a study by Wang, Wang, Zhang, and colleagues regarding the novel role of an acetylesterase and acetyltransferase that acetylate and deacetylate the hypervirulent Klebsiella pneumoniae (Kp) capsule (CPS), the result of which is the modulation of the characteristic hypermucoviscous phenotype. This study builds off of previous work that demonstrated that ompR is required for hypermucoviscosity. Here, the authors show that a small ompR regulon is associated with hypermucoviscosity and narrow this phenotype to a yet uncharacterized gene which they call "KpACE." Overexpression of this gene reduces virulence, and the authors extensively interrogate its potential enzymatic function as an acetylesterase. Correspondingly, they identify an additional acetyltransferase, WcsU, that potentially acetylates CPS, which they also show is necessary for complete virulence. The manuscript is well-written, the data are presented clearly, and I commend the authors for their deep exploration of this interesting phenotype. That said, I feel the authors stopped short of comprehensively demonstrating the association between their genotypes and phenotypes of interest and presenting the strongest possible study.

Reviewer #3: In the manuscript “A novel esterase regulates Klebsiella pneumoniae hypermucoviscosity and virulence,” Wang et. al have identified and characterized a novel esterase (KpACE) that is regulated by OmpR and influences virulence-associated phenotypes and virulence in a mouse model. They determined that this esterase acts on CPS, removing an acetyl group. When overexpressed, this activity was linked to reduced HMV (an important virulence factor), decreased serum resistance, decreased biofilm production, and reduced virulence in mice. The use of mutants that lost activity in multiple assays strengthen the argument that it is deacetylase activity that led to experimental outcomes. In addition, they provide evidence that a putative acyltransferase encoded in the cps locus appears to be the enzyme that adds the acetyl group to CPS. Strains lacking this gene (wcsU) had similar phenotypes to a strain overexpressing KpACE. Collectively, they conclude that acetylated CPS leads to higher HMV, and thus, increased virulence. As the mechanisms driving HMV are not well characterized, these findings provide insight on a new element contributing to this phenotype that will be beneficial to the Klebsiella field. For the most part the assays are well-controlled, the data are convincing, and the manuscript is well-written.

**Part II – Major Issues: Key Experiments Required for Acceptance**

Reviewer #1: The role of at least one of the upregulated genes, 4943, does not appear to be further investigated. Does it make the same amount of capsule (acetylated or not)?

Line 109: The infections in mice would seem to need much for introduction. In fact, the word “mice” does not even appear in this paragraph.

The fact that the overexpression of KpACE results in significantly more capsular polysaccharide (UA in Figure 1H) does not appear to be taken into account in any of the subsequent assays or analyses.

The length of the polysaccharide never seems to be considered.

A double mutant of ∆wbbO and ∆wcsU does not seem to be tested (Figure 3). And a ∆wbbO and pKpACE strain was not tested.

The amount of acetylation does not seem to be tested with the KpACE strain (Figure 3C).

The number of mice in these experiments in Figure 6 is not presented.

It is completely unclear what is being shown in Figure S1. Where are the log2FC values shown?

Figure S6, why should the protein in the supernatant be larger than that in the pellets.

Reviewer #2: 1. The authors generated several useful tools (clean KOs of kpACE and wcsU, kpACE, sub-component, and enzymatically dead kpACE overexpression strains) but they are only selectively used. It would greatly enhance the confidence in the authors' conclusions if they used these tools more comprehensively across the study. For example, why did the authors use the kpACE overexpression strain in their lung infection model but not the clean KO? Why did the authors not explore acetyltransferase activity in serum resistance in Figure 5? As a result of their selective use of these strains, the authors fail to fulfill the molecular Koch's postulates, and as such, the strength of the authors' conclusions is limited.

2. The third catalytic triad (H291) is not described or explored in any detail throughout the study. The data concerning the other triads are convincing, despite being a little confusing regarding the H370 triad-binding phenotype shown in Figure 4C, yet this last triad is ignored throughout the manuscript.

3. Please explain how KO and other strains were confirmed. Ideally, this would be by whole genome sequence, as the capsule locus is highly susceptible to mutations that can confound experimental phenotypes (see PMID 37610214).

Reviewer #3: Major Concerns

1. One particular concern stems from the observation that KpACE reduces HMV in several K2 strains. The strain from which the K2 capsule structure was solved lacked detection of an acetyl group (strain Kp52.145, Ref. 24), but is still HMV positive. This point was not addressed and deserves mention in the discussion. Regarding the strains used in Fig. 3B, it would be very useful to know their origin and whether they contain an intact wcsU gene. The loss of HMV following treatment with KpACE cannot fully be attributed to deacetylation because their acetylation status is unknown. I recognize it is unreasonable to perform NMR on all isolates, so extrapolating their acetylation status from presence/absence of wcsU would be very informative and necessary to conclude that the reduction in HMV is due to deacetylation.

That 52.145 is apparently non-acetylated but still HMV, combined with the lack of acetylation information on the K2 collection leaves open the possibility that KpACE has a second function in addition to deacetylation that contributes to reduced HMV. I would not expect a full exploration of this, but the authors should address the caveats of the data.

2. There are a few conclusions that are overstated. I generally agree with the interpretations, but some wording implies the data are more convincing than they really are.

Some examples:

Line 97: If trying to say that 3347-3349 form an operon, then the wording needs to be softened. Their close proximity on the genome is suggestive on an operon. Use of the word “indicated” led me to think this had been demonstrated experimentally and that was not done.

Line 307: Particularly in light of concerns expressed above, in the sentence “This regulatory cycle is pivotal…”, use of pivotal is a bit overstated. Either modify with a qualifier (e.g. data suggest that this cycle is important) or limit this conclusion to strain 43816.

3. Inclusion of the ∆wcsU mutant in the serum killing and biofilm assays experiments would greatly strengthen their argument for the role of acetylation in these phenotypes.

4. Two supplemental figures lack important controls

Fig. S2. There is no control for non-specific binding. I sought out Wang, et.al 2023 (Ref. 16) and did not see controls there either, but they did have fragments to which OmpR did not find. However, the units of concentration are different so could not extrapolate to the data herein. Please add some information as to the validity of this being specific binding.

Fig. S6. There are lots of reasons that proteins can show up in multiple fractions that have nothing to do with their localization. Please include control antibodies against a protein that is known to not be secreted. The absence of this protein in the supernatant fraction will be more convincing that KpACE is indeed secreted.

**Part III – Minor Issues: Editorial and Data Presentation Modifications**

Reviewer #1: Line 32: This sentence seems like it needs to be switched around and the word “respectively” included.

Reviewer #2: 1. The authors should include negative controls for qPCR assays (e.g., a gene not in the OmpR regulon).

2. In Figure S2, a negative control (e.g., a gene not in the OmpR regulon) would strengthen the authors' findings.

3. In Figure S3, the authors claim that KpACE is highly conserved yet they only interrogate a handful of strains. Given that there are thousands of publicly available Kp genomes, a more broad interrogation would strengthen the authors' conclusions.

4. The gel in Figure S6 is not high quality. Given that there is a band of roughly equal density connecting lanes 3 and 4, one questions if there was spillover loading the gel that impacts the authors' findings.

5. Line 214 should read "harbors."

6. Please explain why only one sex was used in the animal studies and if this should be considered a potential confounder.

7. Regarding the discrepancy in the acetylation of CPS in citations 26 and 27, what do the authors think is the cause of the different observations?

8. Do the authors think that their phenotype is limited to K2 strains?

Reviewer #3: Minor Concerns

1. Although not essential, it would be very interesting to know if KpACE could reduce HMV from other K types such as K1 or others that are associated with HMV.

2. I did not see any information in Material & Methods describing how the cultures were grown with recombinant KpACE (Fig. 3A).

3. The sentence that begins on line 266 “The loss of activity…” is very confusing.

4. Line 154, a sentence suggests that a mutation (His 180), rather than the mutant protein, has retained its activity.

5. Line 109, this sentence reads as if the bacteria had higher survival rates, rather than the mice.

6. Please consider larger fonts on all your graphs. They are very hard to read!

7. Is the RNA-seq data in Table S1 separate from that in Ref. 16? If so, please expand the first paragraph of the results to more comprehensively explain this, and provide information on how to access the raw data (if required by journal). If it’s the same data set, that needs to be indicated.

PLOS authors have the option to publish the peer review history of their article (what does this mean?). If published, this will include your full peer review and any attached files.

Reviewer #1: No

Reviewer #2: **Yes: **Jay Vornhagen

Reviewer #3: No
---

## [Decision Letter · Decision Letter 1]

23 Aug 2024

Dear Professor Wu,

Thank you very much for submitting your revised manuscript "A novel esterase regulates Klebsiella pneumoniae hypermucoviscosity and virulence" for consideration at PLOS Pathogens. Your manuscript was again reviewed by members of the editorial board and by three independent reviewers. In light of the reviews (below), you still need to address several major issues, so we would like to again invite the resubmission of a substantially revised version that takes into account the reviewers' comments.

Please address the two issues raised by reviewer 2. You would ideally provide data on the KpACE deletion strain to address issue 1. Alternatively, please edit the entire manuscript to limit the significance/novelty of the findings and align them with "conclude that artificial repression of mucoviscosity via KpACE reduces infectivity, rather than KpACE being a virulence factor." You should provide data to address issue 2. Alternatively, add the limitation that the observations may be only true for female mice.

We cannot make any decision about publication until we have seen the revised manuscript and your response to the reviewers' comments. Your revised manuscript is also likely to be sent to reviewers for further evaluation.

Sincerely,

Vincent T Lee

Academic Editor

PLOS Pathogens

D. Scott Samuels

Section Editor

PLOS Pathogens

Michael Malim

Editor-in-Chief

PLOS Pathogens

orcid.org/0000-0002-7699-2064

Reviewer's Responses to Questions

**Part I - Summary**

Reviewer #1: The authors have adequately addressed my previous concerns.

Reviewer #2: This manuscript is a resubmission of a study by Wang, Wang, and Zhang. The authors have addressed several, but not all of my concerns. Th inclusion of data supporting a more universal prevalence of KpACE enhances the impact of this study, and the additional controls improves the confidence of the authors findings. Yet, the authors' have yet to fulfill the molecular postulates, and potential sex biases of their findings quench the excitement of the new data thye have provided.

Reviewer #3: In this revised manuscript (PPATHOGENS-D-24-00763-R1), the authors appear to have addressed comments satisfactorily. It is a much smoother read and the additional data and controls have strengthened the conclusions. I have only a few minor comments for consideration.

**Part II – Major Issues: Key Experiments Required for Acceptance**

Reviewer #1: (No Response)

Reviewer #2: 1. Regarding the use of the KpACE overexpression strain, I understand the reasons for using this strain in general; however, the role of mucoviscosity in lung infection is well-established. Thus, the authors are not substantially advancing our understanding of Kp infection, as this strain is experimentally useful but biologically artificial. In the absence of the clean KO, the authors can only conclude that artificial repression of mucoviscosity via KpACE reduces infectivity, rather than KpACE being a virulence factor. The inclusion of strains of interest in the in vitro approaches strengthens the authors conclusions, but the definition of KpACE as a virulence factor seems to be a keystone finding in this study. Without such evidence, the novelty of the authors findings are diminished.

2. Regarding sex as a biological variable, I am not convinced of the authors' arguments. The majority of infection models now use both sexes to ensure there is not a sex bias in the study findings (see PMC8221458 for a topical review). I appreciate that the referenced studies only included female mice, but that is not sufficient justification to ignore sex as a potential confounder in the present study.

Reviewer #3: None

**Part III – Minor Issues: Editorial and Data Presentation Modifications**

Reviewer #1: (No Response)

Reviewer #2: None

Reviewer #3: Some additional explanation for the new RNA-seq experiment would help distinguish this study from the previous one (Ref. 16). Wang, et al. 2023 also states that the RNA-seq experiment was to determine the” OmpR regulon involved in mucoviscosity”. Looking deeper into the Materials & Methods, it appears different media and growth conditions were used—but why?? I realize few readers will notice or care, but as a fan of both HMV and transcriptional regulation, I dig deep into these details. (Also, in Wang, et al. 2023, media listed as “BAP” but this was never defined—assuming Blood Agar Plates?)

Line 157, did the authors perhaps mean “potential” instead of “potent”?

Lines 286-287, in the last sentence, the highlighting that the interaction between CPS & KpACE affected serum killing is a little bit illogical. Not untrue, but it confused me a bit.

I appreciate the added phylogenetic analysis of KpACE genes, but the high sequence similarity made me wonder if the purified enzyme from this K2 strain could be used to treat other HMV+ Klebsiella strains? Having better addressed in this revised version that the broader application of this data to other K types can only be speculated means this experiment is not essential. However, if expression and purification of this enzyme are not exceedingly laborious, this information would greatly enhance the impact of this study on the field of Klebsiella virulence.

PLOS authors have the option to publish the peer review history of their article (what does this mean?). If published, this will include your full peer review and any attached files.

Reviewer #1: No

Reviewer #2: **Yes: **Jay Vornhagen

Reviewer #3: No
---

## [Editor Report · Decision Letter 2]

30 Sep 2024

Dear Professor Wu,

Thank you very much for submitting your manuscript "A novel esterase regulates Klebsiella pneumoniae hypermucoviscosity and virulence" for consideration at PLOS Pathogens. Your manuscript was reviewed by members of the editorial board. 

We appreciate you addressing the key issues raised by the reviewers by providing new experimental data. There are some minor issues with the presentation of the new data that need be addressed:

1. Infection with male and female mice is not indicated in the legend. This should be added to figure 1 and 6.

2. The results for male and female animals is not shown. This should be added as a supplemental figure.

3. S3 Fig. Effects of KpACE deficiency on virulence. The sex of the mice is not indicated.

Please make these changes and we will accept your manuscript.

Sincerely,

Vincent T Lee

Academic Editor

PLOS Pathogens

D. Scott Samuels

Section Editor

PLOS Pathogens

Michael Malim

Editor-in-Chief

PLOS Pathogens

orcid.org/0000-0002-7699-2064

Figure Files:

Data Requirements:

Reproducibility:

References:

---

## [Editor Report · Decision Letter 3]

18 Oct 2024

Dear Professor Wu,

We are pleased to inform you that your manuscript 'A novel esterase regulates Klebsiella pneumoniae hypermucoviscosity and virulence' has been provisionally accepted for publication in PLOS Pathogens. The manuscript improved tremendously through the review process and we appreciate you making the editorial changes.

Best regards,

Vincent T Lee

Academic Editor

PLOS Pathogens

D. Scott Samuels

Section Editor

PLOS Pathogens

Michael Malim

Editor-in-Chief

PLOS Pathogens

orcid.org/0000-0002-7699-2064

---

## [Editor Report · Acceptance letter]

27 Oct 2024

Dear Professor Wu,

We are delighted to inform you that your manuscript, "A novel esterase regulates Klebsiella pneumoniae hypermucoviscosity and virulence," has been formally accepted for publication in PLOS Pathogens.

Best regards,

Michael Malim

Editor-in-Chief

PLOS Pathogens

orcid.org/0000-0002-7699-2064